# GASS: Geometry-Aware Spherical Sampling for Disentangled Diversity Enhancement in Text-to-Image Generation

Ye Zhu [1][2]   Kaleb S. Newman [2]   Johannes F. Lutzeyer [1]
Adriana Romero-Soriano [3][4][5][6]   Michal Drozdzal [3]   Olga Russakovsky [2]

## Abstract

Despite high semantic alignment, modern text-to-image (T2I) generative models still struggle to synthesize diverse images from a given prompt. In this work, we enhance the T2I diversity through a geometric lens. Unlike most existing methods that rely primarily on entropy-based guidance to increase sample dissimilarity, we introduce **G**eometry-**A**ware **S**pherical **S**ampling (*GASS*) to enhance diversity by explicitly controlling both prompt-dependent and prompt-independent sources of variation. Specifically, we decompose the diversity measure in CLIP embeddings using two orthogonal directions: the text embedding, which captures semantic variation related to the prompt, and an identified orthogonal direction that captures prompt-independent variation (e.g., backgrounds). Based on this decomposition, *GASS* increases the geometric projection spread of generated image embeddings along both axes and guides the T2I sampling process via expanded predictions along the generation trajectory. Our experiments on different frozen T2I backbones (U-Net and DiT, diffusion and flow) and benchmarks demonstrate the effectiveness of disentangled diversity enhancement with minimal impact on image fidelity and semantic alignment [1].

## 1. Introduction

Text-to-Image (T2I) generation has gained tremendous popularity and research attention in recent years, driven by ad-

[1]Laboratoire d'Informatique (LIX), CNRS, École Polytechnique, IPP, France [2]Department of Computer Science, Princeton University, USA [3]FAIR at Meta - Montreal, Canada [4]McGill University, Canada [5]Mila, Quebec AI Institute, Canada [6]Canada CIFAR AI chair. Correspondence to: Ye Zhu <ye.zhu@polytechnique.edu>.

*Proceedings of the 43rd International Conference on Machine Learning*, Seoul, South Korea. PMLR 306, 2026. Copyright 2026 by the author(s).

[1]Code is available at https://github.com/L-YeZhu/GASS_T2I.

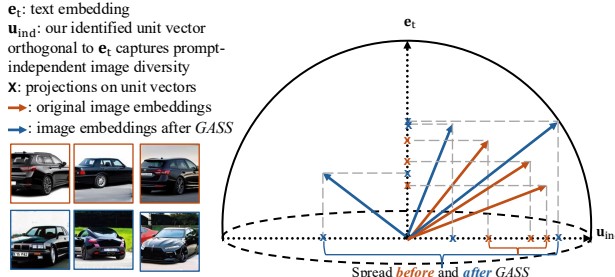

*Figure 1.* **Illustration of our geometric decomposition of sample diversity and *GASS* enhancement method in CLIP space.** We decompose the diversity of generated image batches from T2I models in the CLIP hypersphere along two orthogonal axes: text embedding $\mathbf{e}_t$ (i.e., prompt-dependent) and our identified direction $\mathbf{u}_{\text{ind}}$ (i.e., prompt-independent). Our *GASS* method explicitly expands the geometric spread along both axes, thus enhancing the diversity of generated images across prompt-dependent content (e.g., object viewing angles) and prompt-independent visual attributes (e.g., backgrounds).

vances in model design, including diffusion-based (Ho et al., 2020; Song et al., 2021) and flow-based (Papamakarios et al., 2021; Liu et al., 2023; Lipman et al., 2023) architectures, as well as successful scaling on large-scale text-image datasets (Rombach et al., 2022; Esser et al., 2024). However, despite significant improvements in image fidelity and semantic alignment with text conditions, these models still tend to generate images with limited diversity given a fixed text prompt. This lack of diversity creates practical and societal challenges. It restricts not only the user choice and creative control in generative design workflows, but also risks amplifying societal biases by reinforcing narrow visual stereotypes related to attributes such as gender and ethnicity (Naik & Nushi, 2023; Wan et al., 2024). To address this, we seek to enhance the diversity of generated images within the T2I context under a fixed text prompt in this work.

Prior work often investigates the diversity challenge through the design of evaluation and enhancement methods. From the evaluation perspective, diversity is typically assessed either in a reference-based manner by comparing generated samples against real images using distributional or coverage metrics (Kynkäänniemi et al., 2019; Naeem et al., 2020b),

or in a reference-free manner through quantifying entropy directly in the embedding space (Friedman & Dieng, 2023; Ospanov et al., 2025). As for enhancement techniques, recent methods typically improve diversity by maximizing sample dissimilarity within the batch through perturbations to intermediate latents or conditioning signals (Sadat et al., 2024; Corso et al., 2024; Kirchhof et al., 2025; Ifriqi et al., 2025), aligning with metrics like Vendi Score (VS) (Friedman & Dieng, 2023) that measure embedding entropy. However, these entropy-maximization approaches overlook the multi-sourced nature of T2I diversity. For instance, given a prompt like *"A black colored car"*, outputs vary across prompt-dependent dimensions (e.g., viewing angles, car models) and prompt-independent dimensions (e.g., backgrounds, lighting). While recent Scendi scores attempt decomposition via Schur complement entropy (Ospanov et al., 2025; Jalali et al., 2025a), their reliance on text-image covariance matrices limits applicability to scenarios with equal numbers of prompts and images. Unlike entropy-based approaches, we address this challenge by disentangling and quantifying these sources of variation from a geometric perspective.

We consider the scenario of generating multiple images from a single prompt, and propose to analyze their diversity within the shared CLIP embedding hypersphere (Radford et al., 2021). As illustrated in Fig. 1, we decompose the variation of generated image embeddings $\mathbf{V} = \{\mathbf{e}_i\}_{i=1}^{B}$ relative to the text embedding $\mathbf{e}_t$ into two orthogonal components: the **prompt-dependent variation** captured by their projections onto $\mathbf{e}_t$, which represent semantic changes aligned with the text condition; and the **prompt-independent variation** captured by our identified orthogonal complement $\mathbf{u}_{\text{ind}}$, featuring visual attributes like backgrounds and styles. We further propose to quantify the diversity of the image batch by *summing the respective projection spreads* along each direction. Empirically, we validate this measurement on ImageNet (Deng et al., 2009; Russakovsky et al., 2015) by comparing the geometric spread of real images against synthetic generations from T2I models, as detailed in Sec. 3.

Building upon this geometric analysis, we propose the **G**eometry-**A**ware **S**pherical **S**ampling (*GASS*) method to enhance the generated sample diversity within the T2I setting given a fixed text prompt. Specifically, *GASS* explicitly expands the projection spread of generated embeddings along both orthogonal directions, as illustrated in Fig. 1. We then update the images through gradient-based optimization using the frozen CLIP image encoder. These optimized images are used to replace predicted images within the T2I sampling process, thus steering the generation trajectory toward greater geometric coverage while preserving semantic fidelity. Extensive experiments across diverse T2I backbones (U-Net (Ronneberger et al., 2015) and DiT architectures (Peebles & Xie, 2023), diffusion (Rom-

bach et al., 2022) and flow paradigms (Esser et al., 2024)) and benchmarks (ImageNet (Russakovsky et al., 2015) and DrawBench (Saharia et al., 2022)) demonstrate that *GASS* achieves superior diversity gains compared to state-of-the-art enhancement techniques while maintaining quality and consistency. Notably, to the best of our knowledge, *GASS* is the first sampling-based method to explicitly introduce **meaningful background diversity without modifying text prompts**, as shown in the **non-cherry-picked results** from Fig. 3. This suggests that our geometric formulation enables more comprehensive exploration of the residual space, which has remained largely unexploited in prior work.

Our contributions can be summarized as follows:

- We introduce a geometric framework to disentangle and quantify prompt-dependent and prompt-independent diversity sources within the CLIP hypersphere for T2I generation.

- We propose *GASS*, a geometry-aware spherical sampling method that enhances diversity by explicitly expanding the geometric spread of generated embeddings along orthogonal directions.

- Extensive experiments across diverse T2I backbones and benchmarks demonstrate the effectiveness of *GASS* for disentangled diversity enhancement.

**Conflict of Interest Disclosure.** The authors declare that there are no conflicts of interest.

## 2. Related Work

### 2.1. Diversity Evaluation and Measurement in T2I

Beyond commonly adopted quality and alignment assessment through scores such as FID (Heusel et al., 2017) and CLIPScore (Hessel et al., 2021), sample diversity in T2I remains a critical yet challenging axis of evaluation. Existing assessments can be broadly categorized into reference-based metrics (Kynkäänniemi et al., 2019; Naeem et al., 2020a), which rely on ground-truth data distributions, and reference-free metrics that assess intrinsic sample variety (Friedman & Dieng, 2023; Pasarkar & Dieng, 2024; Jalali et al., 2025b; Ospanov & Farnia, 2024; Ospanov et al., 2024). Specifically, Precision and Recall (Kynkäänniemi et al., 2019), Density and Coverage (Naeem et al., 2020a) are two classic score pairs that simultaneously capture the sample quality and diversity by measuring the distributional overlap between generated samples and real reference data. Image Retrieval Score (IRS) (Dombrowski et al., 2025) is a recently introduced diversity score defined through a retrieval task. In contrast, reference-free metrics assess diversity solely from generated samples. For instance, VS and its recent variants (Friedman & Dieng, 2023; Pasarkar & Dieng, 2024;

Jalali et al., 2025b; Ospanov et al., 2024; 2025) quantify intrinsic diversity via the entropy of the sample similarity matrix. Our work also looks into the diversity in a reference-free manner, by decomposing the batch of image CLIP embeddings into prompt-dependent and prompt-independent components through geometrically grounded orthogonal projection on the high-dimensional unit sphere.

## 2.2. Methods for Enhanced T2I Diversity

Many recent research works aim to enhance generation diversity by introducing additional guidelines in various settings (Ho & Salimans, 2022; Sadat et al., 2024; Miao et al., 2024; Cideron et al., 2024; Corso et al., 2024; Askari Hemmat et al., 2024; Kirchhof et al., 2025; Dall'Asen et al., 2025; Ifriqi et al., 2025; Kynkäänniemi et al., 2024). These efforts can be broadly categorized into post-training-based and inference-time sampling-based approaches. While some works (Miao et al., 2024; Cideron et al., 2024) employ RL-based reward functions during training, a larger body of work (Ho & Salimans, 2022; Sadat et al., 2024; Kirchhof et al., 2025; Kynkäänniemi et al., 2024; Corso et al., 2024) focuses on inference-time guidance. Among the latter, a standard paradigm is to explicitly maximize the entropy of the generated samples (Ifriqi et al., 2025; Corso et al., 2024; Jalali et al., 2025a; Askari Hemmat et al., 2024), directly targeting the information-theoretic definitions of diversity highlighted in Sec. 2.1. While the majority of them share a similar high-level objective of maximizing sample dissimilarity, they often lack granular control over the diversity nature. In contrast, our approach introduces *a layer of controllability*, allowing us to explicitly maximize the diversity along either prompt-dependent (semantic alignment) or prompt-independent (e.g., backgrounds) axes.

## 2.3. Latent Space Analysis for Generative Models

In an orthogonal line of research, recent studies in dynamic generative models (Ho et al., 2020; Song et al., 2021; Liu et al., 2023) have also investigated the geometric structures of the intermediate latent space to unlock more fine-grained control. Based on structural understanding, multiple works (Park et al., 2023; Zhu et al., 2023; Wang et al., 2024; 2025; Baumann et al., 2025) introduce geometrically grounded perturbation and guidance over the sampling process for downstream tasks like image editing and personalization. Despite these advances, the application of such geometric insights to diversity control remains largely underexplored. The most relevant prior work, such as Scendi (Ospanov et al., 2025) and SPARKE (Jalali et al., 2025a), seeks to ground T2I diversity by decomposing the generated images in CLIP space (Radford et al., 2021) with prompt-aware and model-aware components on high-level entropy estimates on distinct text-image pairs. Crucially, metrics like the Scendi Score degenerate to the standard VS

in fixed-prompt settings due to non-invertible covariance matrices, limiting their utility for single-prompt diversity. In contrast, our work leverages explicit geometrical projections to define both a robust way to measure diversity and a corresponding inference-time guidance mechanism.

## 3. Spherically Disentangled Diversity Measure

We now introduce our analysis for disentangling diversity within the CLIP embedding space (Radford et al., 2021). Specifically, we develop a geometric analysis to decompose the variance of a generated batch into distinct prompt-dependent and prompt-independent components, which enables precise measurement of diversity sources.

### 3.1. Motivation and Problem Formulation

Our motivation stems from the inherent under-specification nature of T2I generation: a single text prompt rarely constrains the full semantic and stylistic content of an image. Consider a fixed prompt such as *"A black colored car."* While the prompt explicitly specifies the subject (i.e., the car), it leaves a vast subspace of attributes (e.g., viewing angles, background, etc.) unspecified. This observation suggests that diversity in a generated batch manifests in two distinct forms: variations that adhere to the prompt constraints (prompt-dependent) and variations that explore the unspecified degrees of freedom (prompt-independent).

Based on this intuition, we formalize the problem of diversity measurement as follows: Given a text prompt $c$ and a T2I model $p_\theta$, let $\mathcal{X} = \{\mathbf{x}_i\}_{i=1}^B$ denote a batch of $B$ generated images sampled from the conditional distribution $\mathbf{x} \sim p_\theta(\mathbf{x}|c)$. Our objective is to quantify the diversity of $\mathcal{X}$ not as a single scalar, but as a disentangled tuple $(\mathcal{D}_{\text{dep}}, \mathcal{D}_{\text{ind}})$, where $\mathcal{D}_{\text{dep}}$ (prompt-dependent diversity) measures the variance in how the model interprets the explicit constraints of the prompt $c$; and $\mathcal{D}_{\text{ind}}$ measures the variance of attributes that are orthogonal to the semantic direction defined by $c$. We seek a metric space wherein these sources of variation can be geometrically isolated and measured.

### 3.2. Spherical Disentanglement and Residual Analysis

To achieve the disentanglement formulated above, we require a metric space that satisfies two critical properties: Firstly, it should structurally align visual and textual representations within a shared manifold to allow geometric grounding of image variations relative to the text; Secondly, it should ideally possess a well-behaved topology, such as a hypersphere, to enable non-divergent diversity comparison through geometric constraints. We therefore adopt the $d$-dimensional CLIP embedding space (Radford et al., 2021) for our analysis, where its explicit normalization restricts all embeddings to a high-dimensional unit hypersphere $\mathbb{S}^{d-1}$.

**Algorithm 1** Dominant Residual Basis Identification

---

**Input:** Text embedding $\mathbf{e}_t$, image batch embeddings $\mathcal{P} = \{\mathbf{e}_i\}_{i=1}^B$, number of candidate directions $N$

Generate $N$ direction vectors $\{\mathbf{r}_k\}_{k=1}^N$ orthogonal to $\mathbf{e}_t$ via Gram-Schmidt (Leon et al., 2013).

**for** $k = 1$ to $N$ **do**

$\quad E_k \leftarrow \frac{1}{B} \sum_{i=1}^B |\mathbf{e}_i^\top \mathbf{r}_k|$

**end for**

$k^* \leftarrow \arg\max_k E_k$

**return** $\mathbf{u}_{\text{ind}} \leftarrow \mathbf{r}_{k^*}$

---

Within this spherical geometry, given the normalized text embedding $\mathbf{e}_t$ and a batch of normalized image embeddings $\mathcal{P} = \{\mathbf{e}_i\}_{i=1}^B$. We can strictly decompose each $\mathbf{e}_i$:

$$\mathbf{e}_i = \sum_{k=1}^d \lambda_k \mathbf{u}_k, \ \mathbf{u}_m^\top \mathbf{u}_n = 0 \ \forall m \neq n; \qquad (1)$$

where $\{\mathbf{u}_k\}_{k=1}^d$ forms an orthonormal basis of the embedding space, and $\lambda_k = \mathbf{e}_i^\top \mathbf{u}_k$ are the scalar projection coefficients. By construction, we align the first basis vector with the text prompt (i.e., $\mathbf{u}_1 = \mathbf{e}_t$), so that the first term $\lambda_1 \mathbf{u}_1$ captures the prompt-dependent component, while the remaining terms represent the orthogonal residual. We can therefore rewrite Eq. 1 in CLIP embedding space as follows:

$$\mathbf{e}_i = (\mathbf{e}_i^\top \mathbf{e}_t)\mathbf{e}_t + \sum_{k=2}^d (\mathbf{e}_i^\top \mathbf{u}_k)\mathbf{u}_k. \qquad (2)$$

The scalar projection $\mathbf{e}_i^\top \mathbf{e}_t$ is mathematically equivalent to the CLIPScore (Hessel et al., 2021) for quantifying the semantic consistency between the generated image and the prompt. In theory, a complete characterization of the prompt-independent residual would require analyzing the distribution across the entire $(d-1)$-dimensional orthogonal subspace spanned by $\{\mathbf{u}_k\}_{k=2}^d$. However, in practice, it is known that deep learning representations typically lie on a low-dimensional manifold (Narayanan & Mitter, 2010; Bengio et al., 2013). Consequently, the prompt-independent variation is likely to be not uniformly distributed but highly concentrated along a few principal directions. We therefore simplify the problem by seeking only the *dominant* residual basis vector $\mathbf{u}_{\text{ind}}$ that maximizes the captured variance in the orthogonal complement. We discuss alternative basis construction strategies and provide additional empirical justification for our design choices in Sec. 5.

To identify the optimal residual basis vector $\mathbf{u}_{\text{ind}}$, we employ a randomized search strategy within the tangent space of the text anchor. Specifically, we first generate a candidate set of $N$ direction vectors $\{\mathbf{r}_k\}_{k=1}^N$ that lie strictly within the hyperplane orthogonal to $\mathbf{e}_t$ (i.e., $\mathbf{r}_k^\top \mathbf{e}_t = 0$), and are orthogonal to each other (i.e., $\mathbf{r}_m^\top \mathbf{r}_n = 0 \ \forall m \neq n$) using

*Table 1.* **Quantitative comparison of spherical spread scores (SPP) between generated and real images.** Real images show greater coverage spread than generated ones in both prompt-dependent ($\mathcal{D}_{\text{dep}}$) and prompt-independent ($\mathcal{D}_{\text{ind}}$) measures. Mean and std reported over 1000 classes from ImageNet.

| Img Source | ClipScore | $\mathcal{D}_{\text{dep}}$ | $\mathcal{D}_{\text{ind}}$ | $SPP$ |
|---|---|---|---|---|
| SD2.1 | **0.303**±0.03 | 0.071±0.02 | 0.075±0.02 | 0.146±0.04 |
| SD3-M | 0.302±0.02 | 0.060±0.03 | 0.065±0.03 | 0.126±0.05 |
| Real | 0.293±0.02 | **0.110**±0.03 | **0.110**±0.02 | **0.220**±0.05 |

the Gram-Schmidt orthogonalization (Leon et al., 2013). Ideally, these candidates serve as a representative basis for the high-dimensional residual space. Next, to capture the dominant mode of visual variation unrelated to the text, we evaluate the alignment of the batch embeddings $\mathcal{P}$ with each candidate axis. We compute the mean absolute projection magnitude for each candidate and select the one that maximizes the captured energy:

$$\mathbf{u}_{\text{ind}} = \arg\max_{\mathbf{r} \in \{\mathbf{r}_k\}} \frac{1}{B} \sum_{i=1}^B |\mathbf{e}_i^\top \mathbf{r}|. \qquad (3)$$

This identified vector $\mathbf{u}_{\text{ind}}$ represents the prompt-independent axis that best explains the residual variance of the generated batch. The detailed algorithm is described in Algo. 1. In practice, we empirically observe that a small candidate set size of $N = 10$ is sufficient to robustly identify the principal residual axis, which balances computational efficiency with estimation accuracy.

### 3.3. Spherical Spread Score for Diversity Measure

Having established the orthogonal basis $\mathcal{B} = \{\mathbf{e}_t, \mathbf{u}_{\text{ind}}\}$, we now define a quantitative measure of diversity that captures the dispersion of generated images along these two axes. Specifically, we project the batch of image embeddings $\mathcal{P} = \{\mathbf{e}_i\}_{i=1}^B$ onto each basis vector and quantify the spread of the projected values, yielding two scalar diversity metrics defined as below:

$$\begin{aligned} \mathcal{D}_{\text{dep}} &= \max_i(\mathbf{e}_i^\top \mathbf{e}_t) - \min_i(\mathbf{e}_i^\top \mathbf{e}_t), \\ \mathcal{D}_{\text{ind}} &= \max_i(\mathbf{e}_i^\top \mathbf{u}_{\text{ind}}) - \min_i(\mathbf{e}_i^\top \mathbf{u}_{\text{ind}}). \end{aligned} \qquad (4)$$

$\mathbf{e}_i^\top \mathbf{u}$ stands for projection scalar of $\mathbf{e}_i$ onto the basis vector $\mathbf{u}$. Then we further define our overall diversity spread score as the sum of two spread scores: $SPP = \mathcal{D}_{\text{dep}} + \mathcal{D}_{\text{ind}}$.

Intuitively, these spread scores should effectively distinguish between image sets with varying diversity levels under the same text constraint. To empirically verify this, we compare the spread scores of real images from the ImageNet validation set (Russakovsky et al., 2015; Deng et al., 2009) against samples generated by SD2.1 (Rombach et al., 2022) and SD3-M (Esser et al., 2024) using the template prompt "A photo of *[class label]*". We observe that real-world data,

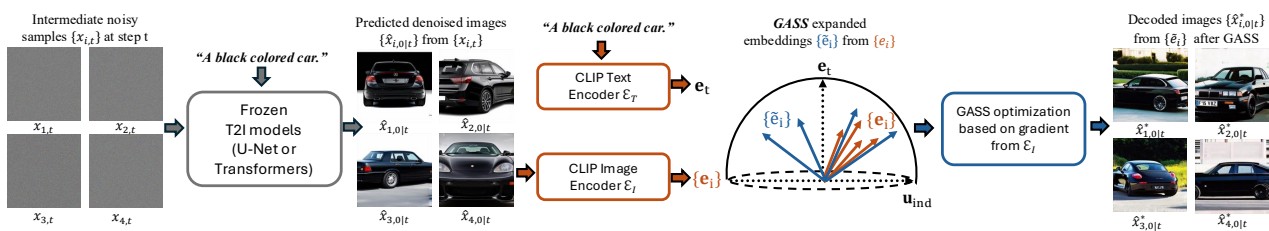

*Figure 2.* **Illustration of our proposed Geometry-Aware Spherical Sampling (GASS) method.** At the generation inference step $t$, original T2I sampling first estimates the predicted clean image $\hat{\mathbf{x}}_{0|t}$ based on the intermediate noisy samples $\mathbf{x}_t$, and then predict the noise we should remove from $\mathbf{x}_t$ to get $\mathbf{x}_{t-1}$. Our *GASS* alters the predicted clean image from $\hat{\mathbf{x}}_{0|t}$ to $\hat{\mathbf{x}}_{0|t}^*$ through the geometric expansion (see Sec. 4.1) and the gradient-based optimization (see Sec. 4.2), thus guiding the iterative sampling process with frozen generative backbones.

---

**Algorithm 2** Optimization based on CLIP Gradient

**Input:** Current batch estimates $\{\hat{x}_{i,0|t}\}_{i=1}^B$, target embeddings $\tilde{\mathcal{P}} = \{\tilde{\mathbf{e}}_i\}_{i=1}^B$, CLIP encoder $\mathcal{E}_I$, step size $\eta$
Encode batch estimates: $\{\mathbf{e}_i\}_{i=1}^B = \mathcal{E}_I(\{\hat{x}_{i,0|t}\}_{i=1}^B)$
$\mathcal{L}_{\text{spp}} = \sum_{i=1}^B (1 - \mathbf{e}_i^\top \tilde{\mathbf{e}}_i)$ {spherical spread loss}
$\hat{x}_{i,0|t}^* \leftarrow \hat{x}_{i,0|t} - \eta \cdot \nabla_{\hat{x}_{i,0|t}} \mathcal{L}_{\text{spp}}$    for $i = 1 \dots B$
**return** $\{\hat{x}_{i,0|t}^*\}_{i=1}^B$

---

which reflects natural complexity, yields significantly higher spread scores (approximately 50% increase) compared to the generated distributions, as detailed in Tab. 1.

## 4. *GASS* for Improved T2I Diversity

Building on our geometric analysis above, we introduce *GASS* (**G**eometry-**A**ware **S**pherical **S**ampling) to intervene in the generation inference process. Formally, we aim to increase the diversity score $SPP$ defined in Sec. 3.3, thereby pushing the newly generated set $\mathcal{X}' = \{x_i'\}_{i=1}^B$ to cover a wider spread of the manifold measured on the CLIP sphere.

### 4.1. Latent Dynamic Spherical Guidance

The first technical challenge is to design a latent perturbation method in the CLIP sphere such that the resulting set of image embeddings $\tilde{\mathcal{P}} = \{\tilde{\mathbf{e}}_i\}_{i=1}^B$ achieves a better spherical spread. Instead of adding isotropic noise in the high-dimensional space like most existing methods (Corso et al., 2024; Kirchhof et al., 2025; Sadat et al., 2024), our geometric framework allows us to inject and control diversity along the disentangled directions.

**Projection Expansion.** For each image $i$ in the batch, we sample an expansion shift $\delta_i^k$ from a uniform distribution:

$$\delta_i^k \sim \mathcal{U}[-r_k, r_k], \qquad (5)$$

where $r_k > 0$ is a hyperparameter controlling the expansion range for the specific axis (i.e., $r_{\text{dep}}$ for prompt-dependent variation along $\mathbf{e}_t$, and $r_{\text{ind}}$ for prompt-independent variation

along $\mathbf{u}_{\text{ind}}$). The perturbed target embedding $\tilde{\mathbf{e}}_i$ is then obtained by modulating the original decomposition:

$$\tilde{\mathbf{e}}_i = (\mathbf{e}_i^\top \mathbf{e}_t + \delta_i^{\text{dep}})\mathbf{e}_t + (\mathbf{e}_i^\top \mathbf{u}_{\text{ind}} + \delta_i^{\text{ind}})\mathbf{u}_{\text{ind}} + \mathbf{r}_i, \quad (6)$$

where $\mathbf{r}_i$ represents the initial residual of $\mathbf{e}_i$ after removing the two principal components in $\mathbf{e}_t$ and $\mathbf{u}_{\text{ind}}$, defined as $\mathbf{r}_i = \mathbf{e}_i - (\mathbf{e}_i^\top \mathbf{e}_t)\mathbf{e}_t - (\mathbf{e}_i^\top \mathbf{u}_{\text{ind}})\mathbf{u}_{\text{ind}}$.

**Re-normalization.** After obtaining the perturbed vector $\tilde{\mathbf{e}}_i$ from Eq. 6, we further apply re-normalization by projecting it back onto the unit hypersphere $\tilde{\mathbf{e}}_i \leftarrow \frac{\tilde{\mathbf{e}}_i}{||\tilde{\mathbf{e}}_i||_2}$. This step ensures that the guided target remains a valid representation within the CLIP embedding manifold, and empirically proven to be beneficial for the generation quality in our experiments in Sec. 5.

**Theoretical Justifications.** Intuitively, our latent spherical guidance directly increases the spread of the image batch in the CLIP embedding manifold. In fact, we can theoretically prove that the expected hypervolume is guaranteed to increase, as detailed below.

**Proposition 4.1** (Expected Geometric Volume Guarantee). *Consider a batch of $B$ points $\mathcal{P} = \{\mathbf{e}_i\}_{i=1}^B \subset \mathbb{S}^{d-1}$ on the CLIP hypersphere, where $\mathbb{S}^{d-1} \subset \mathbb{R}^d$. For each $\mathbf{e}_i$ after our proposed GASS guidance defined in Eq. 6, the new set $\tilde{\mathcal{P}} = \{\tilde{\mathbf{e}}_i\}_{i=1}^B$ has the expected hypervolume $\mathbb{E}[V(\tilde{\mathcal{P}})] > V(\mathcal{P})$.*

The key theoretical insight is that our *GASS* guidance expands the Gram matrix determinant of the point set formed by the batch of images, which translates to the increased geometric hypervolume. A detailed proof of the Proposition 4.1 is included in Appendix A.

### 4.2. SPP Gradient Optimization for T2I Generation

After defining the target embedding set $\tilde{\mathcal{P}} = \{\tilde{\mathbf{e}}_i\}_{i=1}^B$ with enlarged spherical spread, the second technical challenge is to transfer this geometric intervention back into the generative sampling process. Due to the fact that the CLIP does not have pre-trained decoder to directly convert the latent interventions to pixel space, we propose to translate the latent

expansion by leveraging the gradients from the frozen image encoder for guiding the generation. Specifically, at a sampling step $t$, we first estimate the denoised image $\hat{x}_{0|t}$ from the current noisy latent $\mathbf{x}_t$ using the base T2I model's prediction. This estimate is then fed into the CLIP image encoder $\mathcal{E}_I$ to obtain its current batch embedding $\mathbf{e} = \mathcal{E}_I(\hat{x}_{0|t})$. To align the generation with our diversity target $\tilde{\mathbf{e}}$ after *GASS* guidance, we define a batch-wise loss $\mathcal{L}_{\text{SPP}}$ that measures the alignment between the current estimated embedding and the updated target after geometric expansion:

$$\mathcal{L}_{\text{SPP}} = \sum_{i=1}^{B} \left( 1 - \mathcal{E}_I(\hat{x}_{i,0|t})^\top \tilde{\mathbf{e}}_i \right). \tag{7}$$

Crucially, instead of modifying the noise prediction $\epsilon_\theta$, which would require backpropagation through the generative backbone, we directly optimize the estimated clean image $\{\hat{x}_{i,0|t}\}_{i=1}^{B}$. We compute the gradient of the loss and apply a correction step for each sample in the batch:

$$\hat{x}_{i,0|t}^* \leftarrow \hat{x}_{i,0|t} - \eta \cdot \nabla_{\hat{x}_{i,0|t}} \mathcal{L}_{\text{SPP}}, \tag{8}$$

where $\eta$ is the learning rate. This optimized estimate is then substituted into the transition step of existing solvers from pre-trained T2I models, thus effectively steering the generation towards the diverse targets. Detailed optimization algorithm is in Algo. 2. The overall pipeline of *GASS* at a generative sampling step $t$ is illustrated in Fig. 2, with the complete algorithm in Appendix B.

## 5. Experiments

In this section, we describe our experimental setup and present our results and ablation studies [2].

### 5.1. Experimental Setup

**Base T2I Models.** To demonstrate the general applicability of *GASS*, we employ Stable Diffusion 2.1 (SD2.1) and SD3 Medium (SD3-M) (Esser et al., 2024) as our frozen generative backbones. These choices cover a wide spectrum of modern text-to-image generative models, spanning diffusion (Ho et al., 2020) versus rectified flow (Liu et al., 2023) generation paradigms, and U-Net (Ronneberger et al., 2015) versus DiT (Peebles & Xie, 2023) architectures.

**Dataset and Benchmarks.** We evaluate our method on ImageNet-1K (Deng et al., 2009; Russakovsky et al., 2015) and DrawBench (Saharia et al., 2022). For ImageNet, we synthesize 50 images per class using the standard template "A photo of [*class label*]". For DrawBench, we generate 10 samples per prompt and batch. While ImageNet serves as a standard baseline as in prior literature (Kirchhof et al., 2025), DrawBench features prompts with higher semantic

---

[2]All experiments were conducted by LIX and Princeton.

complexity and structural constraints, providing a more rigorous testbed for fine-grained diversity analysis.

**Metrics.** Our evaluation covers sample diversity, generative quality, and semantic consistency alignment. For ImageNet, we employ the classic Density and Coverage (Naeem et al., 2020b) as indicators of fidelity and diversity, respectively. We complement these with ClipScore (Hessel et al., 2021) for alignment, and VS (Friedman & Dieng, 2023) for intrinsic diversity. For DrawBench, due to the absence of reference images, we utilize reference-free metrics: ImageReward (Xu et al., 2023) for perceptual quality, VS for diversity, and ClipScore for consistency. Additionally, we also report our proposed SPP to quantify the geometric spread of the generated samples.

**Diversity Enhancement Baselines.** We compare our approach with four recent and state-of-the-art (SOTA) sampling-based methods designed to enhance sample diversity under fixed prompts: Particle Guidance (PG) (Corso et al., 2024), CADS (Sadat et al., 2024), IG (Kynkäänniemi et al., 2024), and SPELL (Kirchhof et al., 2025). All of these methods are inference-time interventions that amplify diversity by introducing stochastic perturbations to either the intermediate latents or the conditioning signals during the generation sampling trajectory. We replicate all baseline methods, either using their official public implementations or re-implementing them based on the original papers.

**Implementation Details.** To test the generalization ability across varying scales, we generate images at $768^2$ for ImageNet and $512^2$ for Drawbench. During the *GASS* gradient optimization stage, we employ the Adam optimizer with a learning rate of $1 \times 10^{-4}$ for a maximum of 60 steps. We utilize an early stopping strategy with a tolerance of $5 \times 10^{-4}$ and patience of 4 optimization steps. The default inference steps are set to be 50 and 28 for SD2.1 and SD3.5-M, respectively. The default expansion ranges are set to be $r_{\text{dep}} = r_{\text{ind}} = 0.02$. Our proposed *GASS* is a sparse guidance mechanism that can be activated only over a specified interval of sampling steps, reducing computational overhead. For 10∼20 *GASS* guidance steps along the SD3-M generation trajectory, the average cost to sample a batch of images is around 2.93∼3.68 seconds on a Nvidia A100 GPU.

### 5.2. Main Results and Analysis

**Diversity Comparison.** We report the main results on ImageNet and DrawBench in Tab. 2 and Tab. 3, respectively, where we evaluate the generated images in terms of diversity, perceptual quality, and text–image consistency alignment. Compared to recent diversity-enhancement methods that primarily maximize intra-batch sample dissimilarity, our approach improves diversity in both reference-based and

**Methods**           **Non-cherry-picked results from the same batch, each column with same initial noise.**

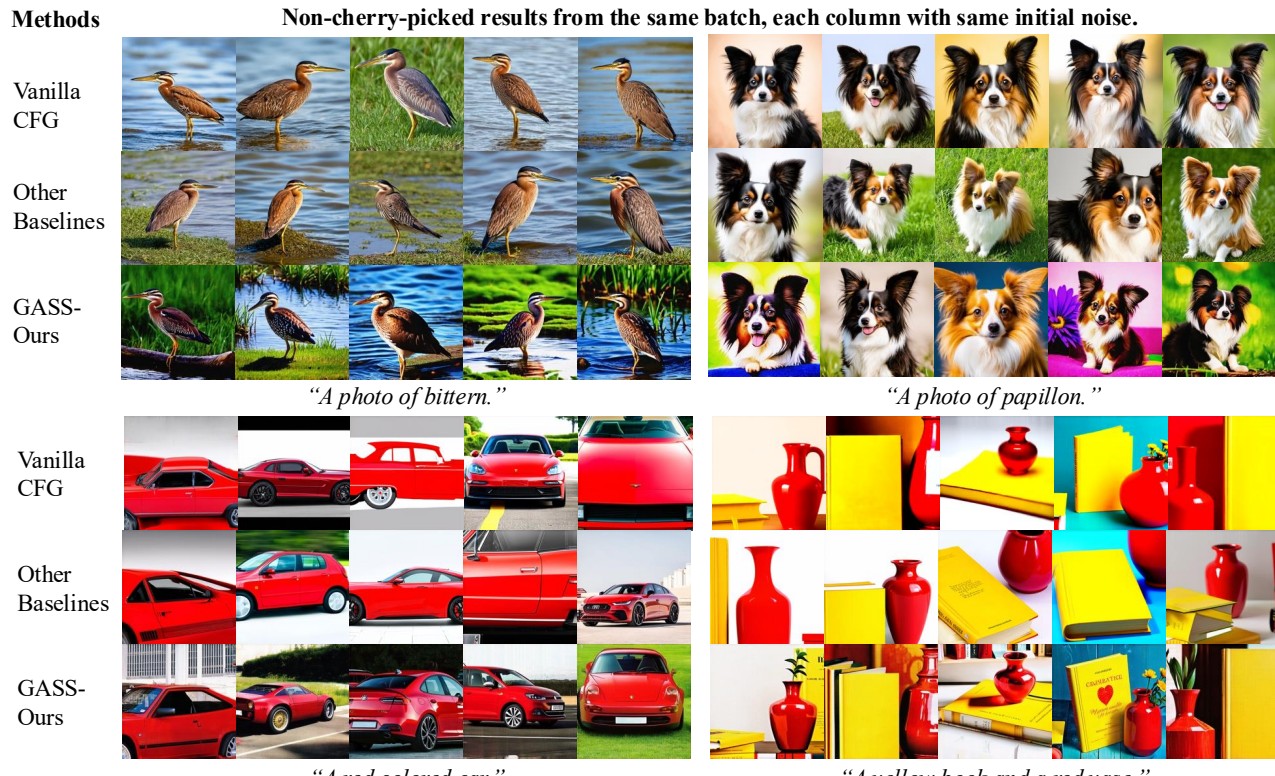

*Figure 3.* **Non-cherry-picked qualitative comparisons with other diversity enhancement methods on ImageNet (Russakovsky et al., 2015) and Drawbench (Saharia et al., 2022).** Compared to other methods (i.e., PG (Corso et al., 2024), CADS (Sadat et al., 2024), IG (Kynkäänniemi et al., 2024), and SPELL (Kirchhof et al., 2025)), our proposed *GASS* generates images with both richer semantic variation (e.g., object poses and layout) and more detailed and diverse backgrounds.

reference-free evaluations, while maintaining competitive quality and consistency. Notably, *GASS* achieves the largest gains on diversity-oriented metrics (e.g., VS (Friedman & Dieng, 2023)) with minimal degradation, or even slight improvements, on quality and alignment metrics, highlighting the effectiveness of our geometry-aware design. This is further qualitatively demonstrated in Fig. 3, where we present **non-cherry-picked** comparisons with baseline methods across both benchmarks. Notably, *GASS* not only introduces semantic variations comparable to other methods but also generates **significantly more detailed backgrounds**. In contrast, other methods often produce ambiguous and smoothed background regions. We attribute this improvement to our explicit expansion of the geometric spread along the prompt-independent orthogonal direction.

**Controllability in Disentangled Diversity Sources.**
Given our orthogonal basis decomposition, we can selectively control diversity enhancement from specific sources by modulating the expansion range $r_k$. In Fig. 4, we illustrate this disentanglement controllability by expanding along the prompt-dependent direction ($\mathbf{e}_t$), prompt-independent direction ($\mathbf{u}_{\text{ind}}$), or both. We empirically observe that prompt-dependent expansion introduces semantic variations such as layout and object poses, while prompt-independent expansion generates diversity through backgrounds and styles.

**Correlation among Diversity, Quality and Alignment.**
Prior work (Zhang et al., 2025; Astolfi et al., 2024) explores evaluation perspectives including diversity, quality, and alignment. Consistent with their findings, Tab. 2 and Tab. 3 reveal this similar trade-off, where diversity gains typically incur quality drops across different methods. In general, our approach achieves superior diversity improvements with minimal quality degradation.

**Diversity under More Complex Prompts.** In addition, it is worth noting that previous works (Ospanov et al., 2025; Zhang et al., 2025) reveal that image diversity can often be introduced through more detailed and specified prompts, thus obscuring the true effect of diversity enhancement from the model perspective. Notably, we demonstrate that *GASS* introduces diversity even under these conditions, as shown in Fig. 5. While vanilla CFG already outputs more diverse images with more specified prompts, our method still introduces further variations across unspecified attributes.

*Table 2.* **Quantitative evaluations on ImageNet with SD2.1 and SD3-M as base T2I models.**

| Base Models | SD2.1 | | | | | SD3-M | | | | |
|---|---|---|---|---|---|---|---|---|---|---|
| Methods | Density↑ | Coverage↑ | VS↑ | ClipScore↑ | SPP↑ | Density↑ | Coverage↑ | VS↑ | ClipScore↑ | SPP↑ |
| Vanilla Guidance (CFG) | **1.130** | **0.623** | 31.826 | 0.308 | 0.143 | 1.105 | 0.588 | 28.119 | 0.308 | 0.137 |
| PG-ICLR'24 | 1.039 | 0.614 | 32.082 | **0.309** | 0.146 | 1.103 | 0.586 | 28.119 | 0.308 | 0.129 |
| CADS-ICLR'24 | 0.897 | 0.567 | 28.935 | 0.306 | 0.142 | 1.374 | **0.636** | 28.456 | 0.309 | 0.133 |
| IG-NeurIPS'24 | 0.901 | 0.574 | 28.935 | 0.306 | 0.140 | **1.389** | 0.627 | 27.415 | 0.310 | 0.129 |
| SPELL-ICML'25 | 0.912 | 0.578 | 32.601 | 0.305 | 0.144 | 1.105 | 0.585 | 28.433 | 0.302 | 0.128 |
| *GASS* (ours) | 1.015 | 0.603 | **32.711** | **0.309** | **0.149** | 1.164 | 0.611 | **28.877** | **0.313** | **0.141** |

*Table 3.* **Quantitative evaluations on DrawBench with SD2.1 and SD3-M as base T2I models.**

| Base Models | SD2.1 | | | | SD3-M | | | |
|---|---|---|---|---|---|---|---|---|
| Methods | VS↑ | ImageReward↑ | ClipScore↑ | SPP↑ | VS↑ | ImageReward↑ | ClipScore↑ | SPP ↑ |
| Vanilla Guidance (CFG) | 8.599 | 0.217 | 0.306 | 0.122 | 8.115 | 0.779 | 0.318 | 0.113 |
| PG-ICLR'24 | 8.637 | **0.231** | 0.306 | 0.121 | 8.002 | **0.799** | 0.318 | 0.113 |
| CADS-ICLR'24 | 8.784 | 0.173 | 0.305 | 0.125 | 8.118 | 0.726 | 0.316 | 0.112 |
| IG-NeurIPS'24 | 8.761 | 0.215 | 0.306 | 0.125 | 8.002 | 0.798 | 0.318 | 0.112 |
| SPELL-ICML'25 | 8.726 | 0.205 | 0.305 | 0.131 | 8.166 | 0.671 | 0.315 | 0.112 |
| *GASS* (ours) | **8.847** | 0.229 | **0.307** | **0.135** | **8.212** | 0.778 | **0.320** | **0.114** |

*Table 4.* **Quantitative analysis w.r.t. different levels of prompt complexities from Drawbench.** We show the performance change compared to the CFG baseline. For instance, 8.301 → 8.535 means that the metric changes from 8.301 (CFG baseline) to 8.535 (with GASS).

| Prompt | VS↑ | ImageReward↑ | ClipScore↑ | SPP↑ |
|---|---|---|---|---|
| Short | 8.301→**8.535** | **0.748**→0.698 | **0.322**→0.321 | 0.113→**0.121** |
| Medium | 7.663→**7.918** | **1.115**→1.043 | 0.318→**0.322** | 0.103→**0.107** |
| Long | 7.549→**7.935** | 0.572→**0.622** | 0.310→0.310 | 0.092→**0.099** |

We further conduct a quantitative analysis. Specifically, we partition the 200 text prompts from the DrawBench (Saharia et al., 2022) into three distinct categories based on word count: short ( ≤ 8 words), medium ( 9 − 14 words), and long/complex ( ≥ 15 words). These categories consist of 92, 62, and 46 prompts, respectively. The qualitative breakdown is provided below in Tab. 4, where we indicate the performance gain compared to the CFG baseline. Interestingly, while human-perceived visual diversity appears to increase with longer and more complex prompts, we observe that diversity metrics (e.g., VS and SPP) actually exhibit a decreasing trend. Despite this, our proposed *GASS* consistently enhances diversity across all prompt complexity categories. Notably, based on the Vendi scores, the margin of improvement achieved by GASS becomes increasingly pronounced as prompt length and complexity grow.

### 5.3. Ablation Studies and Analysis

We further investigate several key designs of the proposed *GASS* method through ablation studies. Tab. 5 and Tab. 6 summarize the quantitative evaluation results.

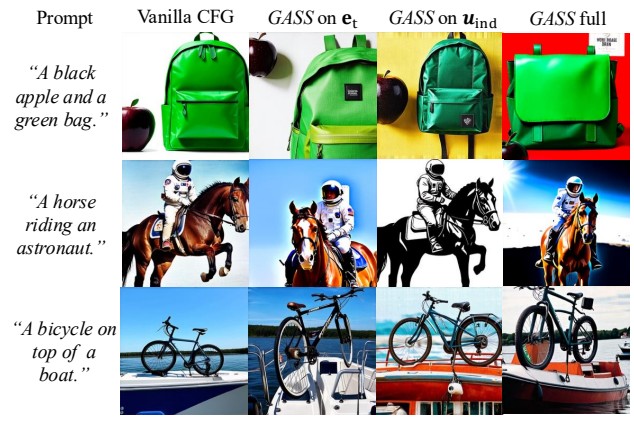

*Figure 4.* **GASS controls the source of diversity by expanding the geometric spread along specified directions.** Specially, *GASS* on prompt-dependent axis $\mathbf{e}_t$ diversifies images through variations via poses and layout, while expansion along prompt-independent direction $\mathbf{u}_{ind}$ changes attributes like background and styles.

**Alternative Decomposition Basis Construction Methods.** To demonstrate the effectiveness of the basis selection method presented in Sec. 3.2, we further investigate two alternative strategies as described below. **Isotropic Perturbation (IP)**: We sample both perturbation directions randomly from an isotropic orthogonal basis; **Random Direction (RD)**: We keep the prompt-dependent base $\mathbf{e}_t$ unchanged, and replace $\mathbf{u}_{ind}$ with a random vector orthogonal to $\mathbf{e}_t$. The results are presented in Tab. 5, where we observe that both isotropic perturbations and a random selection of the dominant residual base on the sphere do not perform well in the T2I diversity enhancement task, demonstrat-

*Table 5.* **Ablation results on different strategies for constructing the decomposition basis.** *IP* stands for Isotropic Perturbation, and *RD* represents Random Direction.

| Method | VS↑ | ImageReward↑ | ClipScore↑ | SPP↑ |
|---|---|---|---|---|
| IP | 8.203 | 0.774 | 0.308 | 0.113 |
| RD | 8.206 | **0.778** | 0.313 | 0.113 |
| *GASS* | **8.212** | **0.778** | **0.320** | **0.114** |

*Table 6.* **Additional ablation results of *GASS* variants.** We show the impact of the re-normalization technique, expansion range, and perturbation steps on Drawbench.

| *GASS* Variants | VS↑ | ImageReward↑ | ClipScore↑ | SPP↑ |
|---|---|---|---|---|
| w/o Norm. | 8.876 | 0.732 | 0.313 | 0.123 |
| $r_{dep} = 0, r_{ind} = 0.02$ | 8.207 | 0.787 | 0.319 | 0.111 |
| $r_{dep} = 0.02, r_{ind} = 0$ | 8.206 | 0.780 | 0.320 | 0.112 |
| $r_{dep} = r_{ind} = 0.02$ | 8.212 | 0.778 | 0.320 | 0.114 |
| $r_{dep} = r_{ind} = 0.05$ | 8.205 | 0.778 | 0.320 | 0.112 |
| t=10 (consecutive) | 8.215 | 0.808 | 0.318 | 0.114 |
| t=10 (uniform) | 8.127 | 0.757 | 0.321 | 0.112 |
| t=15 (uniform) | 8.202 | 0.784 | 0.319 | 0.113 |
| t=20 (uniform) | 8.212 | 0.778 | 0.320 | 0.114 |

ing the effectiveness of our basis construction method. In particular, in the case of an isotropic perturbation, where $e_t$ is replaced by a random direction, the ClipScore drops from 0.320 to 0.308, indicating that it hinders the semantic alignment between the prompt and the generated images.

**Re-Normalization.** In our proposed *GASS* described in Sec. 4, we re-normalize the image embedding $\tilde{e}_i$ to constrain it within the unit hypersphere. Intuitively, this keeps perturbed vectors in the high-density in-distribution region. Our empirical ablation results from Tab. 6 show that removing re-normalization negatively affects the image quality, inducing a drop of ImageReward (Xu et al., 2023) and ClipScore (Hessel et al., 2021) while increasing slightly on diversity measures.

**Expansion Range.** In our proposed sampling method, we define expansion ranges via hyperparameters $r_{dep}$ and $r_{ind}$ along $e_t$ and $u_{ind}$, respectively. Ablation experiments in Tab. 6 show that $r = 0.02$ achieves optimal trade-offs across different evaluation metrics. In addition, expanding along both axes yields the best overall diversity gains compared to single direction expansion.

**Perturbation Steps.** Our *GASS* method is sparse, which requires application only over a subset of sampling steps. We evaluate two scheduler designs, uniform and consecutive *GASS* steps, with results reported in Tab. 6. The early consecutive scheduler applies *GASS* for 10 sampling steps starting from the initial noisy Gaussian; the uniform scheduler steers generation at uniformly sampled steps. The early consecutive scheduler achieves higher VS and ImageReward scores, at the cost of a marginal drop in CLIPScore.

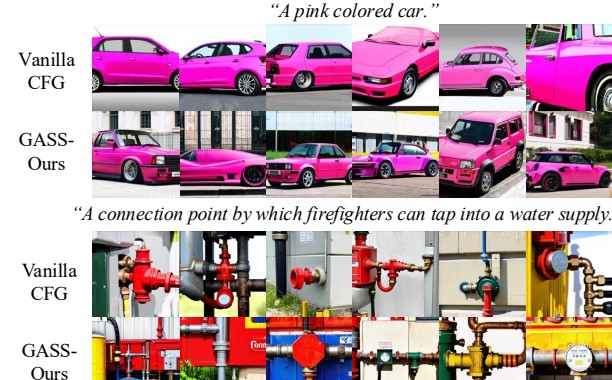

*"A pink colored car."*

Vanilla CFG

GASS-Ours

*"A connection point by which firefighters can tap into a water supply."*

Vanilla CFG

GASS-Ours

*Figure 5.* **GASS still introduces generated image diversity, even when provided with more complex text prompts.**

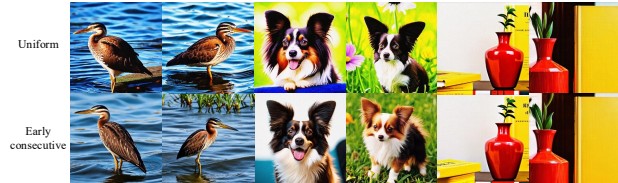

Uniform

Early consecutive

*Figure 6.* **GASS can be applied sparsely on early consecutive or uniform perturbation steps along the T2I generation trajectories.** The early consecutive scheduler tends to generate images with lower color saturation compared to the uniform counterpart.

Qualitatively, consecutive *GASS* steps in the early denoising trajectory tend to generate images with lower color saturation in the final generated images compared to the uniform counterpart, as illustrated in Fig. 6.

## 6. Conclusion and Discussions

In this work, we investigate sample diversity in T2I generation through the lens of spherical geometry. By decomposing diversity into prompt-dependent and prompt-independent components grounded in the geometric structure of the CLIP space, we introduce a principled framework to quantify variation along these orthogonal directions. We further propose *GASS*, a geometry-aware sampling guidance that enhances diversity in a controllable manner via dynamic interventions during inference. Experiments across diverse T2I backbones and benchmarks demonstrate the effectiveness and generalizability of our approach.

A potential future direction could be extending the proposed geometric decomposition beyond prompts (e.g., to multi-condition inputs such as layout or reference images) may enable finer control over which factors of variation are amplified. Limitations and future directions are further discussed in the Appendix D.

## Acknowledgements

This work is supported through the research grant from Meta Inc., under grant number NOA AWD1008796. JL also acknowledges the funding support by the French National Research Agency (ANR) via the "GraspGNNs" JCJC grant (ANR-24-CE23-3888).

## Impact Statement

This paper presents work whose goal is to advance the field of generative models by improving sample diversity in text-to-image generation, with the broader goal of mitigating potential negative societal impacts related to bias and fairness in generated images. There are potential societal consequences of our work, as is similar to the case for many other works in AI generation. However, from a technical standpoint, we do not identify any specific, unusual risks beyond those already associated with contemporary text-to-image generative models.

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

# Appendices

The appendix is structured as follows: First, Sec. A provides the formal theoretical proof of Proposition 4.1 presented in the main paper. Next, Sec. B details the overall algorithm, summarizing the complete process of our proposed *GASS* method. Sec. C includes additional experimental details, including additional implementation details, and more qualitative results. Sec. D discusses the limitations and analyzes failure cases observed in our experiments, proposing several promising future directions.

# A. Theoretical Justification on the Diversity Spread and Hypervolume Expansion after *GASS*

We provide a theoretical justification for the volume expansion induced by our proposed *GASS*, as stated in Proposition 4.1.

**Proposition 4.1** (Expected Geometric Volume Guarantee). *Consider a batch of $B$ points $\mathcal{P} = \{\mathbf{e}_i\}_{i=1}^{B} \subset \mathbb{S}^{d-1}$ on the CLIP hypersphere, where $\mathbb{S}^{d-1} \subset \mathbb{R}^d$. For each $\mathbf{e}_i$ after our proposed* GASS *guidance defined in Eq. 6, the new set $\tilde{\mathcal{P}} = \{\tilde{\mathbf{e}}_i\}_{i=1}^{B}$ has the expected hypervolume $\mathbb{E}[V(\tilde{\mathcal{P}})] > V(\mathcal{P})$.*

*Proof.* The proof proceeds by establishing an explicit relationship between the geometric hypervolume of a point set and the Gram determinant of their inner product matrix. The crucial observation is that independent and orthogonal perturbations through expansion parameter $r_k$ induce positive-definite corrections to the Gram determinant, which increases the hypervolume of the original point set.

**Step 1: Hypervolume via Gram Determinant**

We characterize the hypervolume via the Gram determinant of edge vectors under Gram-Schmidt orthogonalization for the point set $\mathcal{P}$. Specifically, for each pairs of point $\{\mathbf{e}_i, \mathbf{e}_j\}$, we form the edge vector $\mathbf{e}_{ij} = \mathbf{e}_j - \mathbf{e}_i$. We then construct $d$ orthogonal basis as described in Sec. 3.2, including our main expansion axis $\mathbf{e}_t$ and $\mathbf{u}_{\text{ind}}$. By projecting the edge vector $\mathbf{e}_{ij}$ onto $d$ orthogonal basis, and stacking those $B - 1$ linearly independent edge vectors row-wise, we can define the reduced coordinate edge matrix $\mathbf{A} \in \mathbb{R}^{d \times (B-1)}$, the $(B-1)$-dimensional hypervolume is thus given by:

$$V(\mathcal{P}) = \frac{\sqrt{\det(\mathbf{A}^\top \mathbf{A})}}{(B-1)!} = \frac{\sqrt{\det(\mathbf{G})}}{(B-1)!}, \tag{9}$$

where $\mathbf{G} = \mathbf{A}^\top \mathbf{A}$ is the Gram matrix over edge vectors, and $det(\mathbf{A}^\top \mathbf{A})$ is the Gram determinant (Horn & Johnson, 2012).

**Step 2: GASS Expansion on Edge Vectors**

*Part 2.1: Commutativity of projection with perturbations*

We first establish that perturbations applied to projected coordinates are mathematically equivalent to perturbations in the original space followed by projection. Let $\Pi$ denote orthogonal projection onto the $d$-dimensional subspace spanned by orthonormal basis vectors $\{\mathbf{u}_1, \mathbf{u}_2, ..., \mathbf{u}_d\}$.

**Lemma A.1** (Projection Commutativity). *For any vectors $\mathbf{x}$, $\mathbf{y}$ and perturbations $\delta\mathbf{x}$, $\delta\mathbf{y}$, we have:*

$$\Pi(\mathbf{x} + \delta\mathbf{x}) - \Pi(\mathbf{y} + \delta\mathbf{y}) = \Pi(\mathbf{x} - \mathbf{y}) + \Pi(\delta\mathbf{x} - \delta\mathbf{y}). \tag{10}$$

The proof of Lemma A.1 is obvious, because the orthogonal projection is a linear operator, so we have $\Pi(\mathbf{x} + \delta\mathbf{x}) - \Pi(\mathbf{y} + \delta\mathbf{y}) = \Pi(\mathbf{x}) + \Pi(\delta\mathbf{x}) - \Pi(\mathbf{y}) - \Pi(\delta\mathbf{y}) = \Pi(\mathbf{x} - \mathbf{y}) + \Pi(\delta\mathbf{x} - \delta\mathbf{y})$. This ensures that when we perturb the projected coordinates along arbitrary $\mathbf{u}_i$ and $\mathbf{u}_j$, the resulting changes to edge vectors in the projected space directly reflect the perturbations applied.

*Part 2.2: GASS guidance and spread expansion*

Our proposed *GASS* guidance identifies the two orthonormal basis directions $\mathbf{e}_t$, and $\mathbf{u}_{\text{ind}}$ with the largest mean absolute projection scores as specified in Eq. 3. For each point $\mathbf{e}_i$, we apply independent uniform perturbations as described in Eq. 5 and Eq. 6, with:

$$\delta_i^{\text{dep}} \sim \mathcal{U}[-r_{\text{dep}}, r_{\text{dep}}], \delta_i^{\text{ind}} \sim \mathcal{U}[-r_{\text{ind}}, \ r_{\text{ind}}]. \tag{11}$$

The perturbed projected coordinates thus become:

$$\tilde{\alpha}_i = \alpha_i + \delta_i^{\text{dep}}\mathbf{e}_t + \delta_i^{\text{ind}}\mathbf{u}_{\text{ind}}, \tag{12}$$

where $\alpha_i = \Pi(\mathbf{e}_i)$.

*Part 2.3: Impact on edge vectors and Gram matrix*

By Lemma A.1, for any pair of edge vectors, we have:

$$\tilde{\mathbf{e}}_{ij} = \tilde{\alpha}_j - \tilde{\alpha}_i = (\alpha_j - \alpha_i) + (\delta_i^{\text{dep}} - \delta_j^{\text{ind}})\mathbf{e}_t + (\delta_i^{\text{ind}} - \delta_j^{\text{ind}})\mathbf{u}_{\text{ind}}. \tag{13}$$

Let $\mathbf{A} \in \mathbb{R}^{d \times (B-1)}$ be the matrix of original edge vectors, then the perturbed edge vectors form:

$$\tilde{\mathbf{A}} = \mathbf{A} + \Delta\mathbf{A}, \tag{14}$$

where $\Delta\mathbf{A}$ encodes the rank-2 expansion perturbation structure along $\mathbf{e}_t$ and $\mathbf{u}_{\text{ind}}$ introduced by our *GASS* method. The new Gram matrix $\tilde{\mathbf{G}}$ after *GASS* expansion thus become:

$$\tilde{\mathbf{G}} = \tilde{\mathbf{A}}^\top\tilde{\mathbf{A}} = \mathbf{G} + \Delta\mathbf{G}, \tag{15}$$

where $\Delta\mathbf{G} = \mathbf{A}^\top\Delta\mathbf{A} + (\Delta\mathbf{A})^\top\mathbf{A} + (\Delta\mathbf{A})^\top\Delta\mathbf{A}$.

*Part 2.4: Positive-semidefiniteness of the expansion*

*GASS* introduces rank-2 update in the directions $\mathbf{e}_t$ and $\mathbf{u}_{\text{ind}}$ For any vector $\mathbf{v}$, we have:

$$\mathbf{v}^\top(\Delta\mathbf{G})\mathbf{v} = \mathbf{v}^\top(\mathbf{A}^\top\Delta\mathbf{A} + (\Delta\mathbf{A})^\top\mathbf{A} + (\Delta\mathbf{A})^\top\Delta\mathbf{A})\mathbf{v}. \tag{16}$$

The dominant term $|\Delta\mathbf{A}\mathbf{v}|^2 > 0$ is manifestly positive semidefinite. For the rest cross terms that involve the original edge vectors $\mathbf{A}$ and expansions $\Delta\mathbf{A}$, since we align our perturbations with the high-variance basis directions identified by the projection bases as well as empirical justifications, these cross terms preserve non-negativity in expectation, thus we have $\Delta\mathbf{G} \geq 0$.

Part 2.5: Expected volume increase

**Theorem A.2** (Determinant Increase). *If $\mathbf{G}$ is positive definite and $\Delta\mathbf{G} \geq 0$, then we have:*

$$det(\tilde{\mathbf{G}}) = det(\mathbf{G} + \Delta\mathbf{G}) \geq det(\mathbf{G}), \tag{17}$$

*with strict inequality for non-trivial perturbations.*

If $\mathbf{G}$ is positive definite and $\Delta\mathbf{G} \geq 0$, then $\det(\mathbf{G}+\Delta\mathbf{G}) \geq \det(\mathbf{G})$. Intuitively, adding a positive semidefinite perturbation increases (or preserves) all eigenvalues, hence increasing the determinant. A formal proof via eigenvalue perturbation theory is standard; we refer interested readers to linear algebra books (Horn & Johnson, 2012; Strang, 2022).

We can thus derive the hypervolume of the expanded point set $\tilde{\mathcal{P}}$ after *GASS* to be:

$$V(\tilde{\mathcal{P}}) = \frac{\sqrt{\det(\tilde{\mathbf{G}})}}{(B-1)!}. \tag{18}$$

Based on Theorem A.2, we can therefore arrive at:

$$\mathbb{E}[V(\tilde{\mathcal{P}})] \geq \frac{\sqrt{\mathbb{E}[\det(\tilde{\mathbf{G}})]}}{(B-1)!} > \frac{\sqrt{\det(\mathbf{G})}}{(B-1)!} = V(\mathcal{P}). \tag{19}$$

$\square$

## B. Overall Algorithm for *GASS*

In the main paper, we present the algorithms for constructing the spherical basis in Algo. 1, and optimizing image predictions via CLIP-guided gradient updates following *GASS* expansion in Algo. 2. We provide the complete end-to-end algorithm summarizing our *GASS* method in Algo. 3.

---

**Algorithm 3** *GASS* for Diversity Enhancement in T2I

---

**Input:** Text prompt $c$, pre-trained T2I models $p$, CLIP text encoder $\mathcal{E}_T$, CLIP image encoder $\mathcal{E}_I$, number of candidate directions $N$ for dominant residual base construction, expansion ranges $r_{\text{dep}}$ and $r_{\text{ind}}$, step size $\eta$ for optimization, *GASS* sampling step range $\mathcal{T}$.
**Output:** Diverse image batch $\mathcal{X}' = \{\mathbf{x}'_i\}_{i=1}^B$

**Generation Loop**
Encode text prompt: $\mathbf{e}_t \leftarrow \mathcal{E}_T(c)$
Sample random Gaussian initial latent codes $\{\mathbf{z}_{i,T}\}_{i=1}^B$ for generation
**for** $t \in \text{reverse}(\{0, 1, \ldots, T\})$ **do**
  **if** $t \notin \mathcal{T}$ **then**
    **Standard Generation Step:**
    Predict clean latent: $\{\hat{\mathbf{z}}_{i,0|t}\}_{i=1}^B \leftarrow p_\theta(\mathbf{z}_{i,t}, t, \mathbf{e}_t)$    for $i = 1 \ldots B$
    Estimate score and denoise: $\{\mathbf{z}_{i,t-1}\}_{i=1}^B \leftarrow p_\theta(\mathbf{z}_{i,t}, t, \hat{\mathbf{z}}_{i,0|t})$    for $i = 1 \ldots B$
  **else**
    **GASS Optimization Step:**
    *Stage 1: Spherical Decomposition*    (See Sec. 3.2 for details)
    Predict clean latent: $\{\hat{\mathbf{z}}_{i,0|t}\}_{i=1}^B \leftarrow p_\theta(\mathbf{z}_{i,t}, t, \mathbf{e}_t)$    for $i = 1 \ldots B$
    Decode predicted latent: $\{\hat{\mathbf{x}}_{i,0|t}\}_{i=1}^B \leftarrow \text{VAE}_{\text{dec}}(\hat{\mathbf{z}}_{i,0|t})$    for $i = 1 \ldots B$
    Encode images to embedding: $\{\mathbf{e}_{i,0|t}\}_{i=1}^B \leftarrow \mathcal{E}_I(\hat{\mathbf{x}}_{i,0|t})$    for $i = 1 \ldots B$
    Construct orthonormal basis $\{\mathbf{u}_k\}_{k=1}^d$ with $\mathbf{u}_1 = \mathbf{e}_t$ via Gram-Schmidt orthogonalization
    *Stage 2: Residual Basis Identification*
    Generate $N$ candidate direction vectors $\{\mathbf{r}_k\}_{k=1}^N$ orthogonal to $\mathbf{e}_t$
    **for** $k = 1$ to $N$ **do**
      Projection magnitude: $E_k \leftarrow |\mathbf{e}_{i,0|t}^\top \mathbf{r}_k|$
    **end for**
    $k^* \leftarrow \arg\max_k E_k$
    $\mathbf{u}_{\text{ind}} \leftarrow \mathbf{r}_{k^*}$
    *Stage 3: GASS Expansion*    (See Sec. 4.1 for details)
    Compute the residual: $\{\mathbf{r}_i\}_{i=1}^B \leftarrow \mathbf{e}_{i,0|t} - (\mathbf{e}_{i,0|t}^\top \mathbf{e}_t)\mathbf{e}_t - (\mathbf{e}_{i,0|t}^\top \mathbf{u}_{\text{ind}})\mathbf{u}_{\text{ind}}$    for $i = 1 \ldots B$
    Sample perturbations: $\delta_i^{\text{dep}} \sim \text{Uniform}[-r_{\text{dep}}, r_{\text{dep}}]$, $\delta_i^{\text{ind}} \sim \text{Uniform}[-r_{\text{ind}}, r_{\text{ind}}]$    for $i = 1 \ldots B$
    Get expanded image encoding: $\{\tilde{\mathbf{e}}_{i,0|t}\}_{i=1}^B \leftarrow (\mathbf{e}_{i,0|t}^\top \mathbf{e}_t + \delta_i^{\text{dep}})\mathbf{e}_t + (\mathbf{e}_{i,0|t}^\top \mathbf{u}_{\text{ind}} + \delta_i^{\text{ind}})\mathbf{u}_{\text{ind}} + \mathbf{r}_i$,    for $i = 1 \ldots B$
    Re-normalize: $\{\tilde{\mathbf{e}}_{i,0|t}\}_{i=1}^B \leftarrow \tilde{\mathbf{e}}_i / \|\tilde{\mathbf{e}}_i\|$    for $i = 1 \ldots B$
    *Stage 4: GASS Optimization*    (See Sec. 4.2 for details)
    Compute SPP alignment loss: $\mathcal{L}_{SPP} \leftarrow \sum_{i=1}^B (1 - \mathbf{e}_i^\top \tilde{\mathbf{e}}_i)$
    Compute gradient: $\mathbf{g}_i \leftarrow \nabla_{\hat{\mathbf{x}}_{i,0|t}} \mathcal{L}_{\text{SPP}}$    for $i = 1 \ldots B$
    Update predicted latent: $\{\hat{\mathbf{x}}_{i,0|t}^*\}_{i=1}^B \leftarrow \hat{\mathbf{x}}_{i,0|t} - \eta \cdot \mathbf{g}_i$    for $i = 1 \ldots B$
    **Estimate score and denoise:**
    $\{\hat{\mathbf{z}}_{i,0|t}^*\}_{i=1}^B \leftarrow \text{VAE}_{\text{enc}}(\hat{\mathbf{x}}_{i,0|t}^*)$    for $i = 1 \ldots B$
    $\{\mathbf{z}_{i,t-1}\}_{i=1}^B \leftarrow p_\theta(\mathbf{z}_{i,t}, t, \hat{\mathbf{z}}_{i,0|t}^*)$    for $i = 1 \ldots B$
  **end if**
**end for**
Decode final latent: $\mathbf{x}'_i \leftarrow \text{VAE}_{\text{dec}}(\mathbf{z}_{i,0})$ for $i = 1 \ldots B$
**return** Diverse image batch $\mathcal{X}' = \{\mathbf{x}'_i\}_{i=1}^B$

---

# C. Additional Experimental Results

## C.1. Additional Implementation Details

For classifier-free guidance (CFG) (Ho & Salimans, 2022), we adopt the recommended hyperparameter values from the official implementations of each T2I base model. For SD3-M, we set the guidance strength to 5.5 and 7.0 on ImageNet and DrawBench, respectively. For SD2.1, we use a guidance strength of 8.0 on both benchmarks, as recommended values are typically higher than those of SD3-M. For the VS computation (Friedman & Dieng, 2023), we report the VS calculated based on the similarity matrix extracted from the Inception model v3 (Szegedy et al., 2016).

**Particle Guidance (PG)**. PG (Corso et al., 2024) proposes to enhance sample diversity by perturbing the standard sampling process with a potential correction term that converts independent samples into non-i.i.d. samples. For our experiments, we use the publicly available code implementation adapted for both SD3-M and SD2.1, which includes the recommended hyperparameters.

**Interval Guidance (IG)**. IG (Kynkäänniemi et al., 2024) demonstrates that classifier-free guidance can be counterproductive in early denoising steps and redundant in later steps. They propose restricting CFG application to an intermediate interval defined by lower and upper noise level bounds, $\sigma_{\text{lo}}$ and $\sigma_{\text{hi}}$. Following their approach and adapting to models not tested in their work (SD3-M and SD2.1), we perform a grid search over these bounds and select the hyperparameters that yield the best overall performance.

**CADS.** CADs (Sadat et al., 2024) proposes to anneal the conditioning signal by adding monotonically decreasing gaussian noise to the conditioning vector over a fixed interval during the denoising process. CADs has 4 hyperparameters: $\tau_1$, $\tau_2$, noise scale $s$ and a mixing factor $\psi$. For CADs, noise is injected into the conditioning embedding using a linear schedule between $\tau_1$ and $\tau_2$. We fix $\psi$ to 1.0 and fix $\tau_2$ to be more than 1. We further run a grid search over values for $\tau_1$, $\tau_2$, and $s$, we then select the set that lead to the best performance.

**SPELL.** SPELL (Kirchhof et al., 2025) introduces a repellency term that penalizes batch samples whose pairwise distances fall below a pre-defined threshold $r$. Since the original paper does not provide publicly accessible code, we re-implement SPELL based on the provided pseudo-code. Following the original paper, we set the overcompensation coefficient $\lambda = 1.6$ and perform a grid search to determine the optimal radius threshold $r$. We set $r = 250$ for SD2.1 and $r = 350$ for SD3-M.

We summarize the above implementation details and hyperparameter choices from our baseline methods in Tab. 7.

*Table 7.* Hyperparameter settings for different baseline methods across SD2.1 and SD3-M.

| Backbone | Method | Hyperparameters |
|---|---|---|
| SD3-M | IG-NeurIPS'24 | $\sigma \in (0.2, 0.8]$ |
| | PG-ICLR'24 | power coeff = 30.0 |
| | CADS-ICLR'24 | $\tau_1 = 0.9, \tau_2 = 1.20, s = 0.10, \psi = 1.0$ |
| | SPELL-ICML'25 | $\lambda = 1.6, r = 250$ |
| SD2.1 | IG-NeurIPS'24 | $\sigma \in (0.1, 0.9]$ |
| | PG-ICLR'24 | power coeff = 30.0 |
| | CADS-ICLR'24 | $\tau_1 = 0.9, \tau_2 = 1.30, s = 0.15, \psi = 1.0$ |
| | SPELL-ICML'25 | $\lambda = 1.6, r = 350$ |

## C.2. More Qualitative Results

We also include more **non-cherry picked** qualitative results in Fig. 7 and Fig. 8. Consistent with the observations described in the main paper, our proposed *GASS* introduces diversity across multiple dimensions such as semantic-aligned attributes (e.g., viewing angle, object count) as specified in the prompt, as well as unspecified attributes (e.g., background variations).

**Methods**

*"A photo of goldfish."*

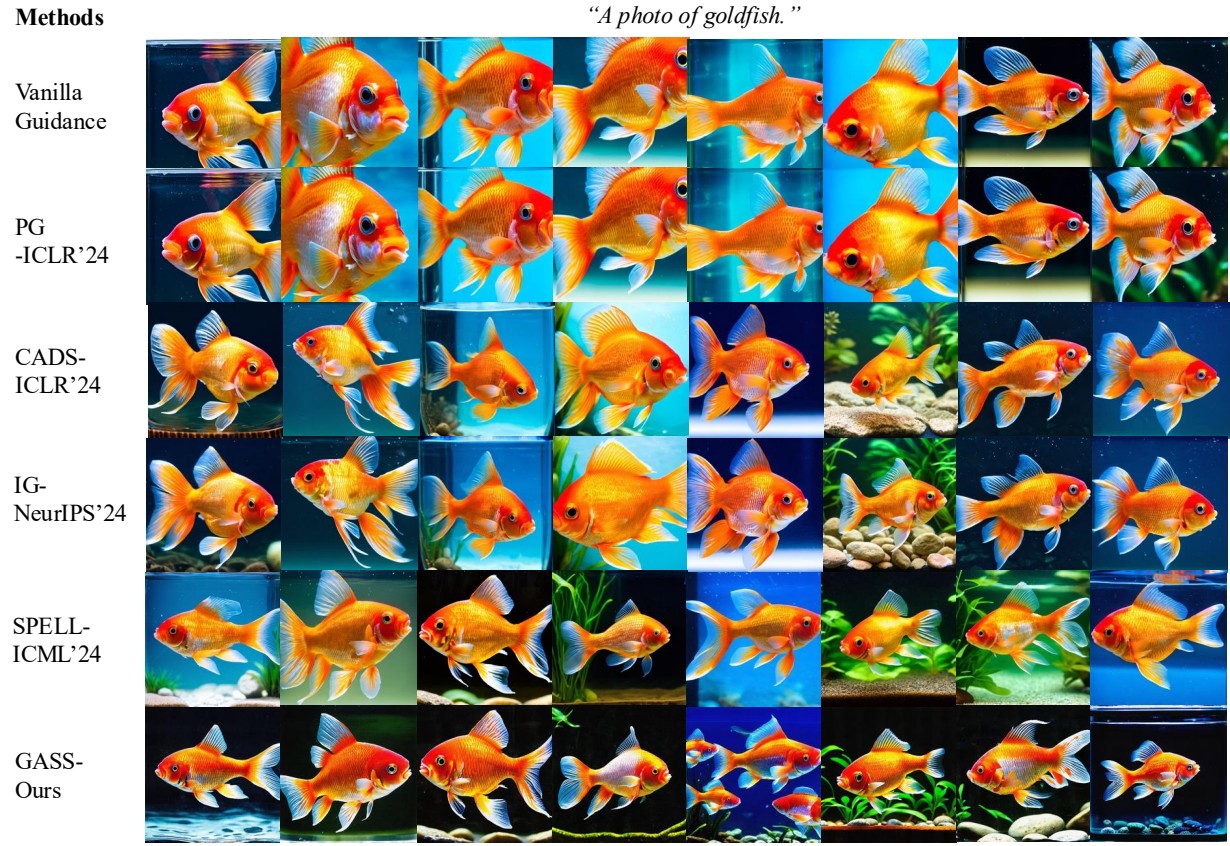

*Figure 7.* **Additional non-cherry-picked qualitative results comparison with other methods on ImageNet with the example class *"goldfish"*.**

## D. Further Discussions

### D.1. Limitations

While our proposed *GASS* method effectively enhances T2I diversity and explores the residual space beyond the given prompts, similar to other sampling-based post-training guidance methods, it incurs extra inference time compared to the original sampling process. Specifically, we note that the major additional computational overhead comes from its current reliance on the CLIP space, which requires extra pixel encoding, gradient-based optimization, and pixel decoding for spread expansion. As we note in the main paper, the current inference time, under 20 *GASS* applied sampling steps, is around 3.68 seconds per batch, versus 1.71 seconds in the original setting.

While *GASS* already offers the possibility to reduce this overhead by applying expansion into fewer steps, one potential future direction to mitigate this is through orthogonal acceleration techniques, such as building a dedicated embedding space directly into the generative model, thus eliminating the need for external CLIP inference. Future work could explore such integrated approaches to achieve geometric diversity guidance with minimal computational overhead.

### D.2. Failure Cases Analysis

While our proposed *GASS* effectively enhances image diversity in most cases, as demonstrated by extensive non-cherry-picked qualitative results in the main paper and appendix, we identify several failure cases where the method produces suboptimal outputs, as illustrated in Fig. 9.

Specifically, we observe that a small fraction of generated images remain similar to baseline samples under vanilla CFG. We attribute this to the fact that our expansion perturbations $\delta_i^{\text{dep}}$ and $\delta_i^{\text{ind}}$ are uniformly sampled from zero-mean distributions.

**Methods**                                        *"A panda making latte art."*

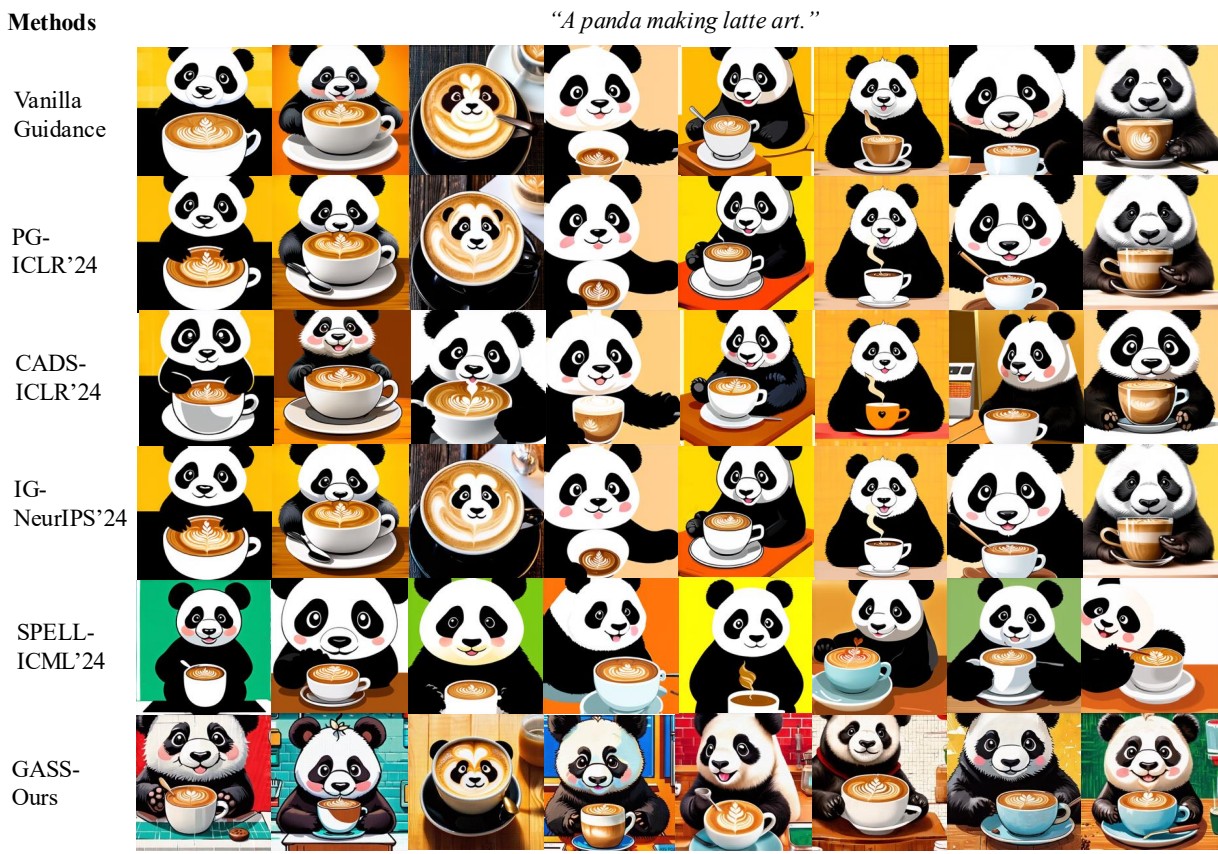

*Figure 8.* **Additional non-cherry-picked qualitative results comparison with other methods on Drawbench.**

While the probability of both perturbations being simultaneously close to zero is very low, such cases do occur, which result in rather trivial modifications after *GASS*. Importantly, these isolated instances **do not impact the overall batch-level diversity metrics**, as diversity is measured across the entire batch rather than individual samples.

**Methods**       *"A photo of cock."*                  *"Three cats and two dogs sitting on the grass."*

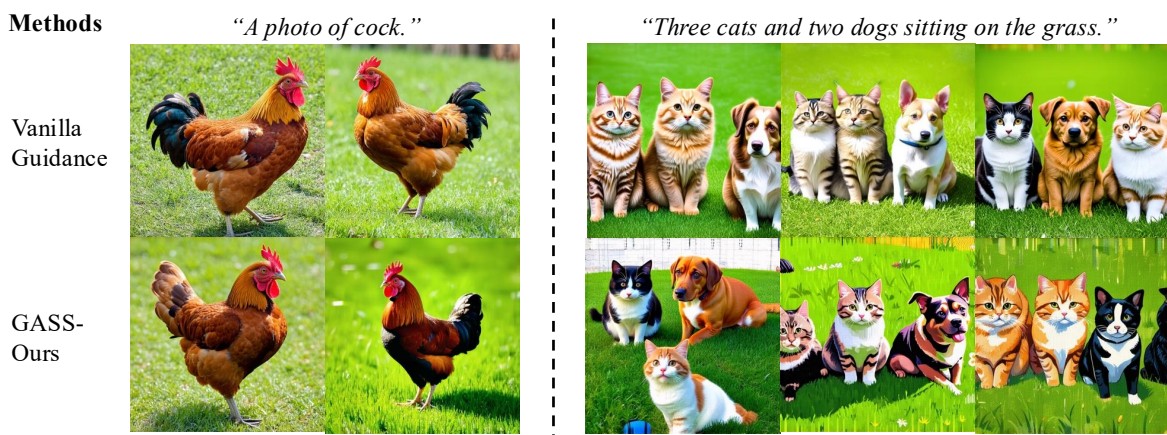

*Figure 9.* **Failure case analysis.** *(Left):* A small number of images still resemble the original ones after GASS. *(Right:)* When the base models can't generate accurate counts specified by the prompt, despite GASS introducing extra diversity, it is less likely to correct these inconsistencies by itself.

In another failure scenario, we observe cases where the base model struggles to accurately follow complex prompts (e.g., "Three cats and two dogs sitting on the grass"). While *GASS* introduces diversity in secondary attributes such as layout and style, it cannot independently correct these semantic misalignments. This limitation stems from our sampling-based approach relying on frozen pretrained models, and thus, we are fundamentally bounded by the base model's capabilities for downstream tasks requiring specific understanding (e.g., numeracy reasoning). Improving performance in such cases would require enhancing the base model itself, which is beyond the scope of post-hoc sampling interventions.

