# OpenReview forum: "GASS: Geometry-Aware Spherical Sampling for Disentangled Diversity Enhancement in Text-to-Image Generation"
_ICML.cc/2026/Conference — ICML 2026 regular_

### Official Review · Reviewer_ggS3 · 2026-03-08

**Soundness:** 3
**Presentation:** 3
**Significance:** 2
**Originality:** 3
**Overall Recommendation:** 3
**Confidence:** 4

**Summary:**

This paper studies improving image generation diversity of a pre-trained text-to-image generation model at test time. Unlike previous methods that use entropy as a measure of diversity, the authors propose to decouple the directions in prompt-dependent and prompt-independent directions. This is done through searching the the linear space that is orthogonal to the original text embedding which aligns the most to explain the variance of the generated images. Next, the iterative denoising processes are modified using the geometric expansion and the gradient-based optimization to increase the diversity.

**Compliance With Llm Reviewing Policy:**

Affirmed.

**Key Questions For Authors:**

What is $v_i^T$ in algorithm 1, should it be $e_i^T$?

**Limitations:**

Yes

**Strengths And Weaknesses:**

The proposed method is simple and relatively novel based on best of my knowledge. The paper is also well-written and easy to follow.

My major concerns about the tackled problem and the proposed method are two-fold:

First, the images generated by the proposed method is visually more saturated and contrast and the original image and baselines. My concern is that by forcing the model to explore the orthogonal directions through optimization, it may favor high-contrast or dramatic visual elements that CLIP perceives as more "distinct" from the original image set. Consequently, this would hurt the image generation quality.

Second, there is a trend that people start to resort to more and more detailed image captions to train the text-to-image diffusion model, which leads to better image generation quality and prompt alignment. As a result, a lot of diversity, for example the background, is captured by the prompts. The diversity in the image generation stage is weakened. Instead, a caption rewriter is used at inference time for better diversity. This makes me concerned about the significance and relevance of the proposed method as this field continue to proceed.

---

> ### Author Rebuttal · Authors · 2026-03-31
>
> We appreciate the insightful comments, and provide our detailed responses supported by additional experimental results below.
>
> ---
> >**W1: First, the images generated by the proposed method is visually more saturated and contrast and the original image and baselines. My concern is that by forcing the model to explore the orthogonal directions through optimization, it may favor high-contrast or dramatic visual elements that CLIP perceives as more "distinct" from the original image set. Consequently, this would hurt the image generation quality.**
>
> **A1:** We thank the reviewer for this insightful observation. We have further investigated this phenomenon and identified an effective empirical solution to mitigate the over-saturation issue by slightly modifying the GASS intervention schedule.
>
> - Specifically, in our original version, the 15–20 GASS steps are **distributed uniformly** across the entire generative trajectory. However, during the rebuttal period, we found that applying GASS **consecutively only in the early generative steps** effectively reduces color saturation while even boosting both the diversity and quality than our uniform guidance scheduler. Furthermore, this early-intervention strategy allows us to reduce the number of GASS guidance steps to just 5 - 10. As shown in the table below, this significantly lowers the computational overhead without compromising the quality and diversity metrics. Qualitative examples demonstrating this mitigated saturation are provided in Figure 2 of [the linked anonymous PDF](https://anonymous.4open.science/r/rebuttal-7C5C/GASS_Figures_rebuttal.pdf).
> |GASS | VS | ImageReward | ClipScore | SPP|
> |------|:--:|:-----------:|:---------:|:---:|
> Uniform 20 steps | 8.212 | 0.778 | **0.320** | **0.114** |
> Early consecutive 10 steps | **8.215** | **0.808** | 0.318 | **0.114** |
>
> - We hypothesize that this approach is effective because it allows the unmodified later generative steps to perform a “post-correction.” Even if the early CLIP guidance pushes the intermediate latent representations toward high-contrast regions, the standard diffusion process in the later steps smoothly refines these trajectories and thus yields more natural final images. We will include this updated intervention strategy and the corresponding analysis in our final version.
>
> ---
>
> >**W2: Second, there is a trend that people start to resort to more and more detailed image captions to train the text-to-image diffusion model, which leads to better image generation quality and prompt alignment. As a result, a lot of diversity, for example the background, is captured by the prompts. The diversity in the image generation stage is weakened. Instead, a caption rewriter is used at inference time for better diversity. This makes me concerned about the significance and relevance of the proposed method as this field continue to proceed.**
>
> **A2:** We appreciate this insightful comment. We would like to address this concern from two perspectives:
> - From a higher-level perspective, we emphasize that the goal of our work is not solely to empirically increase sample diversity, but also to offer a new framework for **understanding T2I models through their implicit latent structures**. This is what motivates our deep dive into their geometric properties. While we acknowledge that prompt rewriting and augmentation via external LLMs is a natural and straightforward way to achieve diversity, it treats the T2I model as a black box and does not improve its interpretability. Our method, by contrast, operates directly on the model’s internal manifold.
> - Interestingly, this trend is partially reflected in our response **A1** to R-pnmw. We provided a detailed quantitative breakdown by parsing the Drawbench prompts based on their length and complexity. This analysis revealed that existing diversity metrics (such as VS) show a counterintuitive trend across different prompt categories. In the meantime, GASS consistently enhances diversity across all categories, regardless of prompt complexity and length. For instance in the table below, 8.301 -> 8.535 means that the metric increases from 8.301 (CFG baseline) to 8.535 (with GASS).
> |Prompt | VS | ImageReward | ClipScore | SPP|
> |------|:--:|:-----------:|:---------:|:---:|
> Short ($\le 8$ words) | 8.301 ->  **8.535** | **0.748** -> 0.698   |  **0.322** -> 0.321 | 0.113 -> **0.121**|
> Medium ($9-14$ words) |  7.663 -> **7.918** |  **1.115** -> 1.043  |  0.318 -> **0.322** | 0.103 -> **0.107**|
> Long ($\ge 15$ words) |  7.549 -> **7.935**  | 0.572 -> **0.622** |   0.310 -> 0.310 | 0.092 -> **0.099**|
>
> ---
>
> > **Q: What is $v_i^T$ in algorithm 1, should it be $e_i^T$?**
>
> **A3:** Thank you for the careful reading. It indeed should be $e_i^T$ as in Equation 3, we will fix this typo in our final version.

---

> > ### Author Rebuttal · Reviewer_ggS3 · 2026-04-04
> >
> > Thank you for your detailed reply and experiments. I appreciate that the new findings that applying GASS in the early denoising timesteps can mitigate the over-saturation issue shown in the paper's results. However, I think this fix has its own limitation. For example, quite of the content generated by the diffusion model is added at a later timestep. Only applying diversification in early steps may lose the power to diversity the mid-frequency or high-frequency content. In addition, I'm concerned that the other aspects may show degradation because of the strong intervention in the early denoising steps. With the limited samples shared, this is hard to determine.

---

> > > ### Author Response · Authors · 2026-04-07
> > >
> > > Thank you for the follow-up question.
> > >
> > > > **However, I think this fix has its own limitation. For example, quite of the content generated by the diffusion model is added at a later timestep. Only applying diversification in early steps may lose the power to diversity the mid-frequency or high-frequency content. In addition, I'm concerned that the other aspects may show degradation because of the strong intervention in the early denoising steps. With the limited samples shared, this is hard to determine.**
> > >
> > > **A1:** We appreciate the careful consideration and address the remaining concerns regarding the early-intervention strategy across the following four aspects:
> > >
> > > 1. **Theoretical Intuitions.** We agree that both diffusion and flow-matching models share a sequential coarse-to-fine generation process. By applying GASS in the early steps, we effectively branch out diverse global content. Due to the continuous sequential nature of the generative ODE/SDE solver, **these structural diversifications naturally propagate along the trajectory**, guiding the model to render correspondingly diverse mid- and high-frequency details in the later steps. Notably, recent sampling-based methods, such as Interval Guidance (IG, NeurIPS’24) [a], also only apply sampling guidance at an intermediate range of steps without alternating the later-stage steps. In fact, one key takeaway from [a] is that they claimed that **“the guidance towards the end is largely unnecessary”**.
> > >
> > > 2. **Quantitative Validation.** This is also quantitatively supported. The early-consecutive guidance strategy yields a higher diversity score (VS) and better general image quality (ImageReward) compared to uniform guidance, avoiding degradation of the concern, while uniform guidance shows slightly better text alignment (CLIPScore). Both GASS variants comfortably outperform recent SOTA methods.
> > > |Method | VS | ImageReward | ClipScore | SPP|
> > > |------|:--:|:-----------:|:---------:|:---:|
> > > |CFG | 8.115 | 0.779 | 0.318 | 0.113 |
> > > |SPELL-ICML’25 | 8.166 | 0.671 | 0.315 | 0.112 |
> > > |GASS - Uniform 20 steps | 8.212 | 0.778 | **0.320** | **0.114** |
> > > |GASS -Early consecutive 10 steps | **8.215** | **0.808** | 0.318 | **0.114** |
> > >
> > > 3. **Extensive Qualitative Evidence.** To address the concern regarding limited visual samples, we have provided more uncurated, non-cherry-picked results on both DrawBench and ImageNet. These qualitative comparisons between the two GASS schedulers also demonstrate that early intervention successfully diversifies the content without losing mid/high-frequency details or causing structural degradation. Please refer to the [anonymous Figure 2 here](https://anonymous.4open.science/r/icml_rebuttal-0447/ICML_GASS_rebuttal2.pdf).
> > >
> > > 4. **Human Preference on Saturation and Quality.** We also note that attributes like color saturation are inherently preference-dependent. To evaluate if early intervention causes perceived degradation, we conducted a pilot human study with 17 CS graduate students. Participants were shown triplets of 10-image sets generated by CFG, uniform GASS, and early-consecutive GASS, and asked to choose their preferred set for a given prompt based strictly on overall visual appeal (independent of diversity). Across 70 triplet comparisons, CFG was preferred 18/70 times, uniform GASS 23/70 times, and GASS with early consecutive scheduling 25/70 times. These results suggest that participants show broadly comparable preferences.
> > >
> > > Finally, we wish to re-state the **overall positioning** of our work. As acknowledged in our original manuscript, the trade-off among diversity, alignment, and visual quality is a longstanding, inherent challenge in generative modeling. Rather than claiming to perfectly eliminate this trade-off, our core contribution is to **provide an interpretable, geometrically-aware, and theoretically grounded sampling framework** (with the formal proof on the expected expansion of the hyper-volume occupied by the generated samples). By calibrating diversity through spherical spread, GASS helps to better disentangle the underlying factors that impact and thus enhance sample diversity. We are happy to further incorporate and emphasize these discussions in the final paper.
> > >
> > > ---
> > > [a] Applying Guidance in a Limited Interval Improves Sample and Distribution Quality in Diffusion Models, NeurIPS 2024.

---

### Official Review · Reviewer_xa16 · 2026-03-09

**Soundness:** 2
**Presentation:** 3
**Significance:** 3
**Originality:** 3
**Overall Recommendation:** 3
**Confidence:** 3

**Summary:**

This paper decomposes the diversity measure in CLIP embeddings into prompt-dependent direction and prompt-independent direction. Based on this decomposition, they enhance diversity by increasing the projection spread of generated image embeddings along these two directions.

**Compliance With Llm Reviewing Policy:**

Affirmed.

**Final Justification:**

I appreciate the authors' effort for clarification.

The authors show that SPP, Scendi Score, and Vendi have the same trends as human preferences. However, they do not show which metric is better aligned with human preferences (e.g., by measuring, over all sample pairs, how often the metric agrees with the human preference). Although the goal of the paper is not to propose a new benchmark metric, the choice of guidance metric for sampling should still depend on which one is best aligned with human preferences for diversity. This work uses the expanded CLIP embeddings to optimize the rgb images, which implicitly use the SPP score as the guidance. It's unclear to me why SPP is used instead of Scendi Score or Vendi if either of them is in fact better aligned with human preferences.  In that case, one could further optimize the computational overhead around the metric that best matches human preferences.

What's more, the 'Early consecutive 10 steps' results show that SPP guidance may be harmful to the color saturation. One possible reason is that, the expanded embeddings used for optimization may no longer lie on the CLIP embedding manifold, leading to the optimized image deviating from the distribution of real images. This seems to be a fundamental limitation of the method, and the claim that it's a common limitation could not convince me.

Therefore, I tend to keep my score.

**Key Questions For Authors:**

My main concerns are Weaknesses 1–5.

**Limitations:**

yes

**Strengths And Weaknesses:**

Strengths

The paper is well motivated, and the decomposition into prompt-dependent and prompt-independent components is intuitive.

Weaknesses

1. The comparison between the proposed SPP score and the Scendi Score (ICCV 2025) [1] is missing. For example, in Table 1, the paper should report both the SPP Score and the Scendi Score for generated images as well as real images, together with a human study, in order to evaluate which metric better aligns with human perception. Similarly, the Scendi Score is also missing from Tables 2 and 3. In addition, it would be better to provide both quantitative and qualitative comparisons by using the Scendi Score as the guidance objective, e.g., by replacing SPP with Scendi in Eq. 8.

2. The comparison with SPARKE (NeurIPS 2025) [2] is missing, both quantitative and qualitative comparisons should be included.

3. Some of the visualized results appear to be over-saturated, and the paper should provide an explanation for this phenomenon. In my view, one possible reason is that, after expansion, the modified vectors may no longer lie on the CLIP embedding manifold. As a result, when these expanded embeddings are used to optimize the RGB image, the optimized image may deviate from the distribution of real images, leading to or over-saturation. If this is indeed the cause, does it suggest a fundamental limitation of the method?
4. In Figure 6 of the appendix, the backgrounds do not appear to be more diverse than those of the baselines.
5. Figure 5 does not include comparisons with the baselines.
6. The paper uses a randomized search strategy with 10 samples to estimate the prompt-independent direction, which may not reliably find the optimal solution. An alternative would be to project the CLIP image embeddings onto the tangent space of the text anchor, and then apply PCA to determine the prompt-independent direction. This would also make it possible to determine the number of prompt-independent axes based on the eigenvalue spectrum, since in some cases a single axis may not be sufficient to capture the full diversity.

[1] Scendi Score: Prompt-Aware Diversity Evaluation via Schur Complement of CLIP Embeddings

[2] SPARKE: Scalable Prompt-Aware Diversity Guidance in Diffusion Models via RKE Score

---

> ### Author Rebuttal · Authors · 2026-03-31
>
> We thank the reviewer for careful and valuable feedback and provide our responses below.
>
> ---
> > **Q1: The comparison between the proposed SPP and the Scendi Score (ICCV 2025) is missing.**
>
> **A1:** We thank the valuable suggestion.
> - We discussed the connection to the Scendi Score (ICCV’25) and SPARKE (NeurIPS’25) in line 64-68 and line 151-160 in our original manuscript. Our decision not to include a **direct quantitative comparison** with Scendi in the initial eval. was based on *two main reasons*: First, our paper focuses on **methodology (a novel SPP-guided sampling technique) rather than benchmarking metrics** as Scendi does. We don’t claim SPP aligns better with human preference; rather, it serves as a lightweight and theoretically grounded proxy specifically designed for inference-time guidance. Second, Scendi and SPP address fundamentally different settings (lines 64-68). Scendi evaluates covariance across diverse text-image pairs (an **$N$-to-$N$ mapping**), whereas we tackle intra-prompt diversity, generating multiple images from a single prompt (a **$1$-to-$N$ mapping**).
> - Nevertheless, to address the concern, we adapted the Scendi to our single-prompt setting using the official repo by repeating the same prompt for all generated images. The updated Tab. 1 below includes the Scendi. Both metrics indicate the same trend for different image sources.
> |ImgSource | SPP | Scendi |
> |------|:--:|:-----------:|
> SD2.1 | 0.146  | 13.608  |
> SD3-M | 0.126 | 10.210 |
> Real | 0.220 |  28.777  |
> - We also evaluated the main experiments in Tab. 2 and 3 with Scendi and observed consistent trends with other diversity measures; however, due to limited space for rebuttal, we will include these in the final version. For instance, SD3-M with CFG achieves Scendi scores of 10.210 and 3.776 on ImageNet and Drawbench, respectively, while for GASS, the values increase to 12.169 and 4.106.
> - Regarding your request to use Scendi in Eq. 8, we explored this variant but faced two major challenges.  First, methodologically, the core idea of GASS is a geometry-aware formulation based on SPP. Substituting it with Scendi, an entropy-based metric, breaks this geometric intuition and is orthogonal to our core contribution.  Second, practically, Scendi incurs a prohibitively high computational overhead. Computing Scendi and backpropagating through it to update the predicted clean images pushes inference time to over 3 minutes per batch in our case, making it practically unviable as an inference-time guidance objective.
> ---
> >**Q2: The comparison with SPARKE (NeurIPS 2025) is missing, both quantitative and qualitative comparisons should be included.**
>
> **A2:** We conducted additional comparisons with SPARKE and will include the comprehensive results in our final paper. For instance, when evaluated on DrawBench using SD3, SPARKE achieves 8.157 (VS), 0.783 (ImageReward), 0.319 (CLIPScore), and 0.113 (SPP). For qualitative visual comparisons, please refer to our response **A5** below.
>
> ---
> > **Q3: Some of the visualized results appear to be over-saturated, and the paper should provide an explanation for this phenomenon.**
>
> **A3:** Due to space limit, please refer to our **A1** to R-ggS3.
>
> ---
> >**Q4: In Figure 6 of the appendix, the backgrounds do not appear to be more diverse than those of the baselines.**
>
> **A4:** The qualitative results shown in our paper were not cherry-picked. Instead, the improvement in background diversity is a fairly general phenomenon that can also be observed in many other empirical examples in the paper. Nevertheless, we appreciate this helpful observation and will include additional examples with more clearly visible background diversity in the final version. We also include several ImageNet examples in Figure 3 of [the anonymous PDF](https://anonymous.4open.science/r/rebuttal-7C5C/GASS_Figures_rebuttal.pdf).
>
> ---
> >**Q5: Figure 5 does not include comparisons with the baselines.**
>
> **A5:** We have integrated additional baseline results including SPARKE, which can be found in Figure 4 of [the rebuttal PDF](https://anonymous.4open.science/r/rebuttal-7C5C/GASS_Figures_rebuttal.pdf).
>
> ---
> >**Q6: Suggestion on the PCA to determine the prompt-independent direction; as well as the question of single axis to capture the diversity.**
>
> **A6:** We appreciate this insightful suggestion.
>
> - We agree that PCA is an intuitive alternative for determining the prompt-independent direction ($u_{\text{ind}}$). Following the advice, we implemented the PCA-based projection strategy and evaluated it on DrawBench. However, we observed virtually no improvement in diversity metrics compared to our current randomized search approach, but extra computational overhead.
> - Furthermore, we empirically explored expanding the intervention from a single axis to multiple orthogonal axes. Similarly, we found that multi-axis guidance did not yield noticeable diversity improvements and actually resulted in a slight degradation in image quality.

---

> > ### Author Rebuttal · Reviewer_xa16 · 2026-04-02
> >
> > Thank you for the reply. Some of my concerns have not been addressed.
> > - Q1: For the fixed-prompt setting, one could copy the prompt from 1 to N in Scendi Score or use Vendi Score. Could you evaluate the alignment of different scores (i.e. SPP and Scendi/Vendi) with the human performance? One would choose the score that better aligns with the human performance as the sampling guidance.
> >
> >   Why does the use of Scendi Score as the guidance need a higher computational overhead? In my view, one could use one prompt to generate N images, then copy the prompt from 1 to N to calculate Scendi Score, or use Vendi Score. Both Scendi Score,  Vendi Score, and the proposed method are calculated on N images, and would have similar computational overhead.
> > - Q2: In DrawBench, SPARKE has a better ImageReward. Can you provide the results on ImageNet and  DrawBench with both SD2.1  and SD3-M?
> > - Q3: 'Early consecutive 10 steps' achieves better ImageReward and relieves the over-saturation issue. This suggests that SPP guidance may be harmful to the color saturation. In my view, one possible reason is that, after expansion, the modified vectors may no longer lie on the CLIP embedding manifold. As a result, when these expanded embeddings are used to optimize the RGB image, the optimized image may deviate from the distribution of real images, leading to or over-saturation. This seems to be a fundamental limitation of the method.
> >
> >   What's more, the author reports a strategy that is not used in the final method of the main paper. Therefore,   all the quantitative and qualitative results in the main paper should be updated and provided.
> > - Q5: The results look still more over-saturated than baselines, especially the right case.

---

> > > ### Author Response · Authors · 2026-04-07
> > >
> > > Thank you for further questions.
> > >
> > > >**Evaluate the alignment of different scores with human performance? One would choose the score that better aligns with the human performance as the sampling guidance.**
> > >
> > > **A1:** We re-clarify that our focus is proposing a novel sampling methodology, not a new metric to compete in human alignment benchmarks. Importantly, GASS does not directly optimize the SPP score. In Eq. (7), $L_{SPP}$ merely minimizes the feature distance between the intermediate latent and and the target one after expansion without explicitly calculating SPP, which also partially explains the larger computational overhead when incorporating Scendi (see **A2**).
> > >
> > > Nonetheless, we conducted a pilot human study (17 CS graduate students) during the rebuttal. Participants ranked the diversity of 10-image sets from three sources (Real Images, SD 2.1, SD3-M) for the same prompt. Across 170 triplet cases, Real Images were consistently ranked most diverse (168/170 times), followed by SD 2.1 (152/170), with SD3-M ranked least (158/170). This human preference aligns closely with the quantitative trends shown by SPP, Scendi, and Vendi, demonstrating their inherent consistency with human perception.
> > >
> > > ---
> > > >**Why does Scendi as the guidance have a higher computational overhead?**
> > >
> > > **A2:** The overhead stems from Scendi’s computation and unstable optimization. In our preliminary implementation, based on the public repo by merely repeating the prompt, Scendi computation alone takes 15.008s per batch on an A100 GPU. Unlike SPP, which is intrinsically normalized and stable, optimizing $\hat{x}_{0|t}$ with Scendi is highly unstable. Using our default hyperparameters in Sec. 5.1, optimization failed to converge within 60 steps, resulting in a 192.711s average sampling time for 10-step guidance. Increasing the learning rate ($5 \times 10^{-3}$) caused instability and distorted generations. We acknowledge that this could potentially be mitigated through a more exhaustive hyperparameter search, but falls outside the scope of our geometry-focused GASS framework.
> > >
> > > ---
> > > >**Can you provide the results on ImageNet and DrawBench with both SD2.1 and SD3-M?**
> > >
> > > **A3:** The full quantitative results for SPARKE are shown in [Tab. 1 this PDF](https://anonymous.4open.science/r/icml_rebuttal-0447/ICML_GASS_rebuttal2.pdf). We note that on SD2.1, SPARKE is sensitive to hyperparameters in our preliminary experiments, requiring further tuning which we will include in the final version. On SD3-M, SPARKE achieves a higher ImageReward (0.783) but lower diversity scores across VS, SPP, and Scendi. This naturally showcases the intrinsic trade-off among generative objectives (alignment, quality, diversity).
> > >
> > > ---
> > > >**This suggests that SPP guidance may be harmful to the color saturation, seems to be a fundamental limitation of the method.**
> > >
> > > **A3:**
> > > - **Color saturation is inherently preference-dependent.** Another pilot study with the same 17 participants asked them to strictly choose their preferred 10-image sets (ignoring diversity). Across 70 comparisons, CFG was preferred 18 times, uniform GASS 23 times, and early-consecutive GASS 25 times. This confirms users hold broadly comparable preferences across different saturation levels.
> > >
> > > - **Deviation from the optimal data manifold is a common limitation of sampling-time interventions,** not unique to GASS. Balancing diversity and visual quality is a well-known trade-off. Most post-sampling methods lack constraints to keep perturbed latents within the implicit and intractable high-density data manifold. Thus, interventions risk pushing trajectories away from natural distributions, degrading quality despite increased entropy. To actively mitigate this, GASS incorporates an explicit re-normalization stage (Sec. 4.1, Line 272).
> > >
> > > ---
> > > >**The results look still more over-saturated than baselines, especially the right case.**
> > >
> > > **A4:** In our updated Fig. 5, we strictly retained the exact uncurated samples from the original manuscript to ensure fair, apples-to-apples comparisons against new baselines. To directly address your saturation concern, we provide an additional figure ([Fig. 1, PDF](https://anonymous.4open.science/r/icml_rebuttal-0447/ICML_GASS_rebuttal2.pdf)) featuring new samples generated via GASS with early-consecutive guidance.
> > >
> > > ---
> > > > **All the quantitative and qualitative results in the main paper should be updated and provided.**
> > >
> > > **A5:** We are happy to include the additional results, since this variant could serve **as an alternative catering to subjective user preferences for lower color saturation**. However, this does not necessitate updating all main results. As extensively demonstrated, there is no absolute “best” strategy for diversity enhancement due to intrinsic generative trade-offs. Both of our proposed GASS strategies effectively enhance sample diversity and achieve the desired disentanglement effect, **fully supporting the main contributions claimed in our original submission.**

---

### Official Review · Reviewer_pnmw · 2026-03-11

**Soundness:** 3
**Presentation:** 2
**Significance:** 2
**Originality:** 3
**Overall Recommendation:** 4
**Confidence:** 3

**Summary:**

The paper studies diversity enhancement for fixed-prompt text-to-image generation and proposes GASS, an inference-time guidance method based on a geometric decomposition of diversity in CLIP space. Specifically, the method separates variation into prompt-dependent and prompt-independent parts and expands both during sampling. The approach is conceptually appealing and practically lightweight in the sense that it does not require retraining. Experiments suggest that GASS improves intrinsic diversity metrics across multiple backbones, although the gains on coverage- and reward-based metrics are more mixed.

**Compliance With Llm Reviewing Policy:**

Affirmed.

**Final Justification:**

The quantitative results and the clarifications provided during the rebuttal phase are robust and sufficiently support the paper's claims.

**Key Questions For Authors:**

1.Can the authors disentangle the effect of the proposed geometry from the effect of added inference-time compute?
A compute-matched control—such as random-direction guidance, isotropic perturbation, or another equally expensive non-geometric intervention—would make the empirical attribution much cleaner.

2.Can the authors provide a dedicated quantitative evaluation for complex / long prompts?

**Limitations:**

yes

**Strengths And Weaknesses:**

Pros:

1.The method is relatively practical and reasonably well scoped.
GASS is an inference-time intervention built on frozen models, and the paper evaluates it on both diffusion and flow backbones, as well as U-Net and DiT architectures. This makes the method more appealing from an engineering standpoint than approaches that require retraining or architecture changes.

2.The experiments support a moderate but real conclusion: GASS consistently improves intrinsic/reference-free diversity.

3.The controllability angle is one of the more interesting aspects of the paper.
At least qualitatively, the paper shows that expanding along the prompt-dependent axis tends to affect layout/pose-related factors, while expanding along the prompt-independent axis tends to affect background/style-related attributes.

Cons:

1.The evidence on complex / long prompts is not strong enough yet.
The paper correctly notes that more detailed prompts can themselves increase diversity. However, this section is only qualitative. There is no dedicated quantitative decomposition of $D_{dep}$, $D_{ind}$, or SPP under complex prompts, and no systematic short-vs-long prompt comparison.

2.The extra computation is nontrivial, and the attribution of gains is still unclear.
The paper reports that with 20 intervention steps, GASS takes about 3.68s per batch versus 1.71s in the original setting. The issue is not only overhead; it is also attribution. Because there is no compute-matched control, it remains unclear how much of the gain comes from the geometry-aware design itself, versus simply doing more test-time optimization(inference-time scaling).

---

> ### Author Rebuttal · Authors · 2026-03-31
>
> We appreciate the reviewer’s recognition of the generalization ability, performance gains, and controllability of our work. We thank the reviewer for the insightful questions and provide our detailed responses below.
>
> ---
> > **Q1: Can the authors provide a dedicated quantitative evaluation for complex / long prompts? (Corresponding to the first point raised in Cons)**
>
> **A1:** We appreciate this valuable suggestion. To provide a fine-grained quantitative evaluation, we partitioned the 200 text prompts from the DrawBench into three distinct categories based on word count: short ($\le 8$ words), medium ($9-14$ words), and long/complex ($\ge 15$ words). These categories consist of 92, 62, and 46 prompts, respectively. The qualitative breakdown table is provided below, where we indicate the performance gain compared to CFG baselines. For instance, 8.301 -> 8.535 means that the metric increases from 8.301 (CFG baseline) to 8.535 (with GASS).
> |Prompt | VS | ImageReward | ClipScore | SPP|
> |------|:--:|:-----------:|:---------:|:---:|
> Short | 8.301 ->  **8.535** | **0.748** -> 0.698   |  **0.322** -> 0.321 | 0.113 -> **0.121**|
> Medium |  7.663 -> **7.918** |  **1.115** -> 1.043  |  0.318 -> **0.322** | 0.103 -> **0.107**|
> Long |  7.549 -> **7.935**  | 0.572 -> **0.622** |   0.310 -> 0.310 | 0.092 -> **0.099**|
>
> Based on the fine-grained quantitative evaluation on DrawBench, we observe an interesting phenomenon: while human-perceived visual diversity appears to increase with longer and more complex prompts, diversity metrics (e.g., VS and SPP) actually exhibit a decreasing trend. Despite this, our proposed GASS consistently enhances diversity across all prompt complexity categories. Notably, the margin of improvement ($\Delta$) achieved by GASS becomes increasingly pronounced as prompt length and complexity grow based on VS.
>
> ---
> > **Q2: Can the authors disentangle the effect of the proposed geometry from the effect of added inference-time compute? A compute-matched control, such as random-direction guidance, isotropic perturbation, or another equally expensive non-geometric intervention, would make the empirical attribution much cleaner. (Corresponding to the second point raised in Cons)**
>
> **A2:** Thank you for the insightful question. Following your suggestion, we ran four additional controlled variants on DrawBench to further validate the effectiveness of our geometry-aware strategy:
>
> • Vanilla Inference Scaling (VIS): Increase the sampling steps of SD3M from 27 to 80, yielding longer inference with an average of 4.10 s per batch.
>
> • Random Direction (RD): Replace $u_{ind}$ with a random vector orthogonal to $e_t$, while keeping $e_t$ unchanged.
>
> • Isotropic Perturbation (IP): Sample both perturbation directions randomly from an isotropic orthogonal basis.
>
> • SPELL with extended intervention: Extend SPELL (ICML’25) to more intervention steps by adjusting the radius hyperparameter, resulting in 3.62s per batch.
>
> Method | VS | ImageReward | ClipScore | SPP|
> |------|:--:|:-----------:|:---------:|:---:|
> CFG (27 steps - 1.71s) | 8.115 | 0.779 | 0.318 | 0.113 |
> VIS (80 steps - 4.10s) | 7.655 | **0.880** |  0.309 | 0.102 |
> RD (3.66s) | 8.206 | 0.778 | 0.313 | 0.113 |
> IP (3.42s) | 8.203 | 0.774 | 0.308 | 0.113 |
> SPELL (3.62s) | 8.197 | 0.712 | 0.316 | 0.112  |
> GASS (3.68s) | **8.212** | 0.778 | **0.320** | **0.114** |
>
> While vanilla inference scaling does not help with diversity in this case, we observe that RD and IP improve VS over the CFG baseline, but clearly degrade ImageReward and ClipScore compared to GASS. After similar inference scaling, we still outperform recent SOTA (e.g., SPELL-ICML’25) under a similar sampling time budget. This confirms that **perturbing random directions without geometric structure harms text–image alignment**. In contrast, GASS improves VS while preserving or even slightly improving alignment metrics. We will include these results as an additional ablation in the revised paper.
>
> We will incorporate these clarifications into the final manuscript.

---

> > ### Author Rebuttal · Reviewer_pnmw · 2026-04-02
> >
> > Thank you for the detailed rebuttal. I will maintain the positive score recommendation.

---

### Official Review · Reviewer_MYmY · 2026-03-11

**Soundness:** 4
**Presentation:** 4
**Significance:** 3
**Originality:** 3
**Overall Recommendation:** 4
**Confidence:** 2

**Summary:**

This paper proposes GASS a sampling method for enhancing image diversity of T2I models by expanding the projection spread of generated embeddings via gradient-based optimization in CLIP space.

**Compliance With Llm Reviewing Policy:**

Affirmed.

**Final Justification:**

My concerns and questions were addressed.

**Key Questions For Authors:**

Can the GASS be applied to other generative models? FLUX or DiT?

How performance of GASS on generating diverse human face images?

**Limitations:**

yes

**Strengths And Weaknesses:**

Strengths:

Authors provided a formal poof to verify that GASS increases the expected hypervolume in Proposition 4.1. rather than heuristic findings.

The proposed methods achieved overall better performance than other SOTAs in various evaluation metrics.

Weaknesses:

Relying on CLIP Space: GASS's effectiveness is restricted to the geometry of the CLIP embedding space. Therefore, the diversity performance is bounded to training dataset of CLIP model.

Computational overhead, GASS introduce significant time cost, i.e., 3.68s vs 1.71.

Only two diffusion models are tested which limits generalization of method.

---

> ### Author Rebuttal · Authors · 2026-03-31
>
> We appreciate the reviewer’s recognition of our formal theoretical proof of coverage expansion, as well as the valuable feedback. We provide our responses below.
>
> ---
> > **Q1: Can the GASS be applied to other generative models? FLUX or DiT?**
>
> **A1**: **Yes, GASS can be generalized to a wide range of T2I models.**
> - In fact, our framework is highly compatible with the latest architectures. While classic T2I models (e.g., SD 2.1) rely on U-Nets and DDPMs, our main experiments also evaluate SD3, which is based on the **DiT architecture and Rectified Flow formulation**, is highly similar to the MMDiT and Rectified Flow used in FLUX.
> - Following the suggestion, we conducted additional experiments using FLUX-Schnell on DrawBench. As shown in the table below, GASS achieves consistent performance gains on this new architecture. We restrict this comparison to the vanilla FLUX-Schnell and GASS due to two factors: the high inference latency of FLUX (~29.5s per image), and the incompatibility of baselines like Interval Guidance (IG) with few-step models (IG requires an intermediate range of timesteps, whereas FLUX-Schnell defaults to only 4 steps).
> | Method | VS | ImageReward | ClipScore | SPP |
> |------|:--:|:-----------:|:---------:|:---:|
> |FLUX-Schnell | 7.348 | **1.006** | 0.307| 0.089 |
> |GASS | **7.886** | 0.998 | **0.309** | **0.102** |
>
> ---
> > **Q2: How performance of GASS on generating diverse human face images?**
>
> **A2:** **GASS also enhances diversity for synthesizing human face images.**
> - To answer the question and demonstrate the effectiveness, we use LLMs to generate 100 different prompts to ask for human face generation, with an average length of 13.4 words per prompt, covering different attributes such as age, expressions, and hairstyles. Example prompts include: *“Headshot of an elderly woman with short curly hair, smiling warmly, natural light.”*
> - As this is a reference-free scenario, we compute and report the scores used in Drawbench as follows. Additional non-cherry-picked examples can be found in Figure 1 in [this anonymous link PDF](https://anonymous.4open.science/r/rebuttal-7C5C/GASS_Figures_rebuttal.pdf):
> | Method | VS | ImageReward | ClipScore | SPP |
> |------|:--:|:-----------:|:---------:|:---:|
> |CFG | 7.046 | **1.029** | 0.297 | 0.085|
> |GASS | **7.353** | 0.986 | **0.298** | **0.106** |

---

> > ### Author Rebuttal · Reviewer_MYmY · 2026-04-01
> >
> > Thanks, I will maintain my score.

---

### Decision · Program_Chairs · 2026-04-30

**Decision:**

Accept (regular)

**Comment:**

This paper proposes a geometrically motivated inference-time method for improving text-to-image diversity by decomposing variation into prompt-dependent and prompt-independent components during sampling. Reviewers generally agreed that the problem is interesting and that the method is practically appealing, and several of the empirical weaknesses were addressed in rebuttal through additional controls and analyses. In particular, the controllability perspective and its attempt to offer a more interpretable view of diversity enhancement were viewed as meaningful contributions. The main remaining concern relates to the visual over-saturation observed in some visualization results, the extent to which the proposed early-step intervention resolves this issue. While these concerns are not entirely settled, the overall discussion suggests this submission with solid strengths and empirical support to merit a borderline acceptance recommendation.